# Particle Aging and Aerosol–Radiation Interaction Affect Volcanic Plume Dispersion: Evidence from Raikoke Eruption 2019

Lukas O. Muser[1], Gholam Ali Hoshyaripour[1], Julia Bruckert[1], Akos Horvath[2], Elizaveta Malinina[3], Sandra Wallis[4], Fred J. Prata[5], Alexei Rozanov[3], Christian von Savigny[4], Heike Vogel[1], and Bernhard Vogel[1]

[1]Institute of Meteorology and Climate Research, Karlsruhe Institute of Technology (KIT), Karlsruhe, Germany
[2]Meteorological Institute, University of Hamburg, Germany
[3]Institute of Environmental Physics, University of Bremen, Bremen, Germany
[4]Institute of Physics, Greifswald University, Greifswald, Germay
[5]AIRES Pty. Ltd., Mt Eliza, Victoria, Australia

**Correspondence:** Lukas O. Muser (lukas.muser@kit.edu)

**Abstract.** A correct and reliable forecast of volcanic plume dispersion is vital for aviation safety. This can only be achieved by representing all responsible physical and chemical processes (sources, sinks, and interactions) in the forecast models. The representation of the sources has been enhanced over the last decade, while the sinks and interactions have received less attention. In particular, aerosol dynamic processes and aerosol–radiation interaction are neglected so far. Here we address this gap by further developing the ICON-ART (ICOsahedral Nonhydrostatic – Aerosols and Reactive Trace gases) global modelling system to account for these processes.

We use this extended model for the simulation of volcanic aerosol dispersion after the Raikoke eruption in June 2019. Additionally, we validate the simulation results with measurements from AHI (Advanced Himawari Imager), CALIOP (Cloud-Aerosol Lidar with Orthogonal Polarization), and OMPS-LP (Ozone Mapping and Profiling Suite – Limb Profiler). Our results show that around $50\,\%$ of very fine volcanic ash mass (particles with diameter $d < 30\,\mu m$) is removed due to particle growth and aging. Furthermore, the maximum volcanic cloud top height rises more than $6\,km$ over the course of $4$ days after the eruption due to aerosol–radiation interaction. This is the first direct evidence that shows how cumulative effects of aerosol dynamics and aerosol–radiation interaction lead to a more precise forecast of very fine ash lifetime in volcanic clouds.

## 1 Introduction

Volcanic aerosols pose significant hazards to aviation (Casadevall, 1994; Guffanti et al., 2010; Schmidt et al., 2014), and influence weather and climate (Robock, 2000; Mather, 2008). These aerosols are primarily composed of ash particles (tephra with diameter smaller than $2\,mm$) (Rose and Durant, 2009). Secondary aerosols are generated from precursor gases, such as sulfate particles from $SO_2$, through chemical and microphysical processes (Tabazadeh and Turco, 1993; Textor et al., 2004; Durant et al., 2010).

During the first couple of days after the onset of an eruption, aerosol concentration can be locally so high that it jeopardizes air traffic. In the past, most of the aircraft damaging encounters occurred in spatial proximity ($< 1000\,km$) to the volcano or

within 24 hours after the onset of ash-producing eruptions (Guffanti et al., 2010). In order to provide a timely response to such events, a reliable forecast of volcanic aerosol dispersion is crucial. This is a challenging task because of large uncertainties in dispersion models mainly with respect to the eruption source parameters (e.g., mass eruption rate and plume height) and internal model parameterizations (e.g., wet deposition, aerosol dynamics, and optical properties) (Prata et al., 2019; von Savigny et al., 2020). While the model sensitivities to the source parameters were extensively studied in recent years (e.g. Mastin et al., 2009; Harvey et al., 2018), the role of the aerosol dynamics in plume dispersion remains largely unexplored.

Aerosol dynamic processes comprise nucleation, condensation, coagulation, and sedimentation. These processes alter the aerosol size and composition (particle aging) and thus, modify the optical properties of particles (Seinfeld and Pandis, 2016). Such changes eventually affect the aerosol dispersion and their interactions in the atmosphere (Abdelkader et al., 2017; Peterson et al., 2017; Yu et al., 2019). This can result in a lofting mechanism of aerosol which is different from the one caused by large scale atmospheric dynamics as described for example by Khaykin et al. (2017). Abdelkader et al. (2017) studied the sensitivity of transatlantic dust transport to chemical aging. The results show that chemical aging of dust particles increases the aerosol optical depth under subsaturated conditions and leads to regional radiative feedbacks to surface winds and dust emissions. Besides, the aged dust particles are removed more efficiently (by both wet and dry deposition) due to the increased hygroscopicity and particle size (Abdelkader et al., 2017). Peterson et al. (2017) observed in the Arctic near-surface atmosphere that the transport of atmospheric pollutants is influenced by active halogen chemistry. Yu et al. (2019) used modeling and satellite observations to characterize the effect of particle chemistry on smoke plume lofting after forest fires in Canada in August 2017. They reported that the smoke plume rose from 15 to 20 kilometers within 10 days owing to solar heating of aged black carbon.

Change of particle size during volcanic ash dispersion has been the topic of ash aggregation research in the last three decades (see Brown et al., 2012, and the references therein). Aerosol dynamics is one of the dominant mechanisms than lead to volcanic ash aggregation during long–range transport (Brown et al., 2012). Numerical models only (if at all) consider wet aggregation in the eruption column (Textor et al., 2006; Van Eaton et al., 2015; Folch et al., 2016; Marti et al., 2017). This can lead to an underestimation of the ash fallout and overestimation of airborne ash mass concentrations 1000s km from the volcano (Brown et al., 2012).

Previous works have studied the effects of aerosol–radiation interaction on the ash and $SO_2$ dispersion after historic eruptions assuming externally mixed aerosols (Niemeier et al., 2009; Schmidt et al., 2014). Niemeier et al. (2009) showed that the radiative effect of fine ash particles (strong absorption of shortwave and long-wave radiation) causes additional heating and cooling of $\pm 20$ K per day and modifies the evolution of the volcanic cloud. Such impacts can be substantial in short-term at local scale and strongly depend on the optical properties of the volcanic particles (Niemeier et al., 2009; Timmreck, 2012; Vernier et al., 2016). It has been shown that volcanic ash particles interact and mix with other aerosols (Delmelle et al., 2007; Ayris and Delmelle, 2012; Bagnato et al., 2013; Hoshyaripour et al., 2015). This aging process affects the chemical composition and size distribution of the ash particles and can have a profound impact on their optical properties (Durant et al., 2010; Vogel et al., 2017). It is not clear yet how particle aging affects the dispersion and radiative impacts of volcanic ash. Here, we aim at exploring this gap by extending the ICON-ART (ICOsahedral Nonhydrostatic – Aerosols and Reactive Trace gases) global

modelling system (Zängl et al., 2015; Rieger et al., 2015) by a new aerosol dynamic module named AERODYN (AEROsol DYNamics). This new extension allows us to investigate the formation of secondary aerosols and aerosol aging. In the scope of this paper we focus on timescales on the order of several days after the onset of an eruption. The primary focus is on

the dynamics of the volcanic cloud during this initial period to provide information for volcanic aerosol dispersion forecasts. Therefore, we quantify the influence of secondary aerosol formation and particle aging on the optical properties of the volcanic particles. The research questions are as follows: 1) What is the influence of aerosol dynamics and ash aging on volcanic aerosol dispersion? 2) What is the effect of aerosol–radiation interaction on volcanic aerosol dispersion? 3) Are the representations of aerosol dynamics and aerosol–radiation interaction beneficial for volcanic aerosol dispersion forecast?

To answer these questions we investigate the Raikoke eruption in June 2019. The Raikoke volcano ($48.29°$ N, $153.24°$ E) is a stratovolcano located on Raikoke island, one of the central Kuril islands in the Sea of Okhotsk. An eruption started on June 21, 2019, at 18:00 UTC (Sennet, 2019). The large ash plume rapidly rose to $8–14$ km altitude. A series of nine explosive events occurred until 05:40 UTC on 22 June. Forty airplanes were diverted because of the ash plume produced by this eruption (Sennet, 2019).

The paper is structured as follows: in Sect. 2 we present the observational data used in this study. Furthermore, the ICON-ART modeling system is described together with the simulation setup. Section 3 presents the results and the discussion of the very. Answers to the posed research questions are given in Sect. 4.

## 2 Methodology

### 2.1 Observation data

#### 2.1.1 $SO_2$ from TROPOMI

The spread of the $SO_2$ plume ejected by the Raikoke eruption in June 2019 as well as the amount of released $SO_2$ mass was investigated by analyzing $SO_2$ total vertical column densities from the hyperspectral nadir-viewing TROPOspheric Monitoring Instrument (TROPOMI) aboard the Sentinel-5 Precursor satellite. TROPOMI provides daily global coverage completing $14.5$ orbits every day (van Kempen et al., 2019) with a pixel size of $7$ km $\times$ $3.5$ km (Theys et al., 2019). TROPOMI $SO_2$ (daylight

only) offline level 2 data were downloaded from the Copernicus website (https://s5phub.copernicus.eu). The total vertical $SO_2$ column densities used, assume a $SO_2$ profile described by a $1$ km thick box at $15$ km altitude to account for explosive volcanic eruptions (Theys et al., 2017).

A self-defined geographic grid including the area from $30°$ N $–$ $75°$ N and $135°$ E $–$ $120°$ W with a resolution of $0.1° \times 0.1°$ was created. The $SO_2$ cloud expansion for every TROPOMI orbit was visualized by first averaging all vertical $SO_2$ column

densities inside a single grid segment and multiplying the result by the $SO_2$ molar mass in order to obtain a mass loading in units of $\mathrm{g\,m^{-2}}$. Only data with a quality value larger than $0.5$ (as recommended in the TROPOMI product user manual) and total vertical column density with values less than $1000$ $\mathrm{mol\,m^{-2}}$ were used.

The $SO_2$ mass loading for each grid segment was multiplied subsequently with the associated grid segment area to obtain the $SO_2$ mass in units of g. The total $SO_2$ mass for the observed area was determined for the observed area over time periods of approximately $24$ h, i.e., by averaging batches of 14 consecutive orbits for every single grid segment. Finally, the mass is summed up over the entire grid. The described data averaging was applied because consecutive orbits partially overlap. This method suggests a total emitted $SO_2$ mass of $(1.37 \pm 0.07) \times 10^9$ kg over the course of the Raikoke eruption 2019. Since the air mass factor used in the retrieval of the vertical column densities depends on the $SO_2$ vertical distribution, the choice of the assumed $SO_2$ profiles seems to be the most important source of error. It remains, however, a non-trivial challenge to estimate the associated uncertainty of the $SO_2$ mass calculation. The uncertainty stated above reflects the average absolute difference between the $SO_2$ mass calculated from an assumed $SO_2$ profile peak in $15$ km and $7$ km altitude, respectively. $SO_2$ masses from 20 June, 16:41 UTC to 6 July, 10:08 UTC were included in the averaging.

### 2.1.2 Ash and $SO_2$ from Himawari-8

Himawari-8 is a geostationary satellite platform operated by the Japanese Space Agency (JAXA) in collaboration with the Japanese Meteorological Agency (JMA) carrying the 16 band visible and infrared Advanced Himawari Imager (AHI). Data are acquired every 10 minutes over the Earth's disc covering a circular field of view of approximately 70 degrees, centred at the equator and $\sim 140°$ E longitude. Further details of the orbit, instrument, duty cycles, image geolocation, and data calibration can be found on the JAXA/JMA website and in documentation (https://www.data.jma.go.jp/mscweb/en/himawari89/space_segment/spsg_ahi.html).

For the purpose of this work, AHI infrared data were analysed at $10$ min intervals to determine the column amounts of $SO_2$ gas and ash particle mass loadings, both in units of $g\,m^{-2}$. At the sub-satellite point the nominal spatial resolution of infrared pixels is $4$ km$^2$, increasing to $> 100$ km$^2$ at the largest scan angles. The Raikoke plume covered a relatively large geographic region and range of latitudes/longitudes, so the data were first rectified and resampled to a grid of $1336 \times 2139$ latitude $\times$ longitudes centred at $52.5°$ N latitude and $175°$ E longitude using a stereographic projection. These infrared data were then processed to determine $SO_2$ and ash amounts at $10$ min intervals. The final data were analyzed at both $10$ min and hourly intervals. The basis of the retrieval of $SO_2$ slant column amount relies on using AHI band 10 centred near to $7.3$ µm. At this wavelength there is a strong $SO_2$ absorption band. Water vapor and clouds cause interference with the $SO_2$ signal and introduce a positive bias. Therefore, a retrieval scheme was devised to minimize the interfering effects. In short, the bias is minimized by subtracting an offset $SO_2$ retrieval for a small region where no $SO_2$ is believed to exist. Details of the retrieval method are very similar to a scheme devised for the High Resolution Infrared Sounder (HIRS) data described by Prata et al. (2003).

Volcanic ash effective particle radius and optical depth are retrieved using AHI bands 14 ($\sim 11.2$ µm) and 15 ($\sim 12.4$ µm) on the same latitude/longitude grid as that used for $SO_2$. The basic physics has been described by Prata (1989) and the retrieval methodology has been described by Prata and Prata (2012) using Meteosat Second Generation (MSG) Spin-Enhanced Visible and Infrared Imager (SEVIRI) data, which has very similar characteristics to the AHI data used here.

Discussions of potential error sources in ash retrieval can be found in numerous papers in the literature, e.g., Wen and Rose (1994); Prata et al. (2001); Clarisse et al. (2010); Mackie and Watson (2014); Western et al. (2015). Prata and Prata (2012) and Clarisse and Prata (2016) provide some error estimates based on independent validation which suggest single pixel retrievals have an absolute error of $\pm 0.5 \, \mathrm{g \, m^{-2}}$ with a low bias; however, much larger errors and biases can occur on occasion and it is generally accepted that relative errors typically lie between 40–60 %. Single pixel retrievals $< 0.2 \, \mathrm{g \, m^{-2}}$ are regarded as at the threshold of detection. The presence of ice reduces the ash mass estimates by an amount that depends on the proportion of the pixel covered by ice. However, during the Raikoke eruption, ice was not observed except possibly at the start of the eruption which could cause lower ash mass estimates.

The retrieval assumes that pixels detected as containing ash are completely ash covered and although meteorological cloud tests are used, inevitably some anomalous retrievals occur. To minimise these, a mask was used whereby all pixels falling outside a $0.1 \, \mathrm{g \, m^{-2}}$ contour line are removed. Within the $0.1 \, \mathrm{g \, m^{-2}}$ contour, a $9 \times 9$ median filter was applied to remove any remaining "spikes". These measures are largely cosmetic and are based on the premise that anomalous pixels appear to be unphysical in nature. Integrating the horizontal mass loadings for volcanic ash and $SO_2$ their emitted masses can be estimated. Based on the AHI measurements the total emitted very fine ash mass ($d < 32 \, \mu m$) ranges between $0.4$–$1.8 \times 10^9 \, \mathrm{kg}$, the $SO_2$ mass between $1$–$2 \times 10^9 \, \mathrm{kg}$. The latter agrees well with the TROPOMI measurement in Sect. 2.1.1.

### 2.1.3 Volcanic cloud height from MODIS, VIIRS, OMPS, and CALIOP

There are several ways of obtaining volcanic cloud top heights in the upper troposphere and lower stratosphere. In this work, we use data from four spaceborne instruments, MODIS (Moderate Resolution Imaging Spectroradiometer), VIIRS (Visible Infrared Imager Radiometer Suite), OMPS-LP (Ozone Mapping and Profiling Suite – Limb Profiler), and CALIOP (Cloud-Aerosol Lidar with Orthogonal Polarization). These instruments are briefly described in the following.

We used meteorological cloud top height (CTH) and volcanic ash cloud top height retrievals from MODIS aboard the Terra and Aqua satellites and VIIRS aboard the S-NPP (Suomi National Polar-orbiting Partnership) and NOAA-20 satellites. These polar–orbiting instruments observed Raikoke on 22 June 2019, at 01:25 UTC (Terra and NOAA-20), 02:15 UTC (S-NPP), and 03:10 UTC (Aqua and NOAA-20) when a brownish–colored and still localized plume was largely distinguishable from white/gray meteorological clouds in visible channel images. MODIS cloud top height, available at $1 \, \mathrm{km}$ horizontal resolution, is obtained by matching the retrieved cloud top pressure to a Numerical Weather Prediction (NWP) geopotential height profile (Menzel et al., 2015). For the Raikoke plume, classified essentially as ice phase with a few liquid phase pixels, cloud top pressure was mostly determined by the $CO_2$–slicing technique from channels near $13 \, \mu m$ and to a lesser degree by the infrared window technique from the $11 \, \mu m$ channel. For VIIRS, on the other hand, cloud top height was determined only from the $8.5$, $11$, and $12 \, \mu m$ channels, because the instrument lacks $CO_2$ absorbing channels. The NOAA Enterprise AWG (Algorithm Working Group) Cloud Height Algorithm (ACHA) first determines cloud top temperature (CTT) from these midwave infrared channels using an optimal estimation framework and then matches CTT to a collocated NWP temperature profile (Heidinger and Li, 2019). The VIIRS CTHs are available at $750 \, \mathrm{m}$ horizontal resolution. In addition to the meteorological cloud products, VIIRS retrievals by a dedicated volcanic ash detection and height algorithm (Pavolonis et al., 2013) were also utilized. The

optimal estimation method is based on the same midwave infrared channels as used in the cloud retrievals, but the underlying microphysical models assume particles (andesite, quartz, kaolinite, or gypsum) that are better suited for volcanic plumes than liquid water or ice. A series of spectral and spatial tests first select only those pixels that potentially contain volcanic ash, which makes retrieval coverage more restricted compared to the standard cloud product, especially in scenes containing a mix of ash and water clouds. The algorithm then retrieves ash cloud effective temperature and effective emissivity, from which ash cloud height is computed with the help of NWP temperature profiles. The estimated ash height error was typically 1–2 km for the Raikoke plume. Despite their different assumptions about plume microphysics, the cloud and ash height retrievals agreed well where both were produced and indicated a maximum plume top height between 12–12.6 km about 8 h after the start of the eruption.

The volcanic cloud top height on 22 June 2019, was determined by visual analysis of the stratospheric aerosol extinction coefficient profiles from the OMPS-LP instrument. Here, the aerosol extinction coefficient product at 869 nm (V1.0.9) retrieved at the University of Bremen is used. The OMPS aerosol extinction coefficient was retrieved on a 1 km grid from 10.5 to 33.5 km with the algorithm adapted from the SCIAMACHY V1.4 (Rieger et al., 2018). The retrieval is done under the assumption that stratospheric aerosol is represented by spherical sulfuric droplets with a unimodal log–normal particle size distribution ($r_{med} = 80$ nm, $\sigma = 1.6$). Due to uncertainties in pointing and vertical sampling we estimate the measurement error with $\pm 0.7$ km. Detailed information on the retrieval algorithm can be found in Malinina (2019) and Malinina et al. (2020). Here, it should be noted that the evaluation of the plume top height from OMPS-LP was possible only on the 22 June 2019. On that day, the instrument was passing right above the Raikoke island, and the plume was very localized. Thus, the increase in the aerosol extinction coefficient associated with the eruption was large and obvious. This large increase was a result of a vast amount of ash released with the eruption. In the following days, when the plume started to spread over the North Pacific, the core of the fresh plume is not hit by the OMPS-LP instrument sampling anymore. Slightly perturbed aerosol extinction observed in transition regions has a similar magnitude as that from interfering events, e.g., the aerosol transport from the Ambae eruption that occurred 11 months earlier, and thus cannot be attributed exclusively to the Raikoke eruption. For this reason, we excluded the OMPS-LP measurements in transition regions from the consideration.

CALIOP is one of three instruments on board the CALIPSO (Cloud-Aerosol Lidar and Infrared Pathfinder Satellite Observation) satellite, which was launched on 28 April 2006 and is still operational. CALIOP provides backscatter measurements at 532 nm and 1064 nm and the backscattered radiation at 532 nm is measured in two channels detecting orthogonally polarized radiation. The determination of the Raikoke plume height is based on total attenuated backscatter data at a wavelength of 532 nm. CALIOP L1 data version 4.10 is used.

In the scope of this paper, we analyze the CALIPSO overpass on 23 June 2019, at around 15:00 UTC. On this date the total attenuated backscatter at 532 nm shows a distinct feature between 15 and 16 km that can be associated with the volcanic cloud.

## 2.2 Modeling system and set up

### 2.2.1 ICON-ART modeling system

This study uses the ICOsahedral Nonhydrostatic weather and climate model with Aerosols and Reactive Trace gases (ICON-ART). ICON is a non-hydrostatic modeling system that solves the full three-dimensional non-hydrostatic and compressible Navier–Stokes equations on an icosahedral grid (Zängl et al., 2015). ICON can be used for seamless simulations of various processes across local to global scales (Heinze et al., 2017; Giorgetta et al., 2018). The ART module is an extension of ICON to account for emission, transport, physicochemical transformation, and removal of the trace gases and aerosols in the troposphere and stratosphere (Rieger et al., 2015). Zängl et al. (2015), Rieger et al. (2015), and Schröter et al. (2018) provide detailed technical descriptions of ICON and ICON-ART, respectively. The removal of aerosols from the atmosphere is modeled by three different processes: sedimentation, dry deposition and wet deposition. In ICON-ART wet deposition describes scavenging by raindrops below clouds.

The Rapid Radiative Transfer Model (RRTM) (Mlawer et al., 1997) is used in ICON as the standard radiation scheme for numerical weather prediction. To account for the aerosol radiative effect, ART calculates the local radiative transfer parameters (extinction coefficient, single scattering albedo, and asymmetry parameter) based on the optical properties and the prognostic mass concentration of aerosols at every grid point and for every level. These are then used as the input parameters for the RRTM scheme (Gasch et al., 2017). This approach ensures full coupling and feedback between aerosol processes, radiation and the atmospheric state (Shao et al., 2011). Besides, a forward operator is implemented in the model to diagnose the attenuated backscatter at the wavelengths $532$ and $1064\,\text{nm}$ (Hoshyaripour et al., 2019). To account for secondary aerosol formation and internally mixed aerosols, a new aerosol dynamics module is currently developed and implemented in ICON-ART. Details of this module are described in the following section.

### 2.2.2 Aerosol dynamics

The aerosol dynamics module (AERODYN) includes 10 log–normal modes that consider Aitken, accumulation and coarse particles in soluble, insoluble and mixed states plus a giant insoluble mode. This new development allows a very flexible combination of different species for different ICON-ART applications. The Aitken, accumulation, coarse (in all mixing states) and giant modes are initialized with geometric median diameter of $0.01$, $0.2$, $2.0$ and $12.0\,\mu\text{m}$ and standard deviations of $1.7$, $2.0$, $2.2$ and $2.0$, respectively. Figure 1 provides additional information about the organization of the modes and species in AERODYN.

For each mode prognostic equations for the number density and the mass concentration are solved while the standard deviations are kept constant. The generalized aerosol dynamics equations have the following form:

$$\frac{\partial}{\partial t} M_{0,i} = -Ca_{0,ii} - Ca_{0,ij} + Nu_0 \tag{1}$$

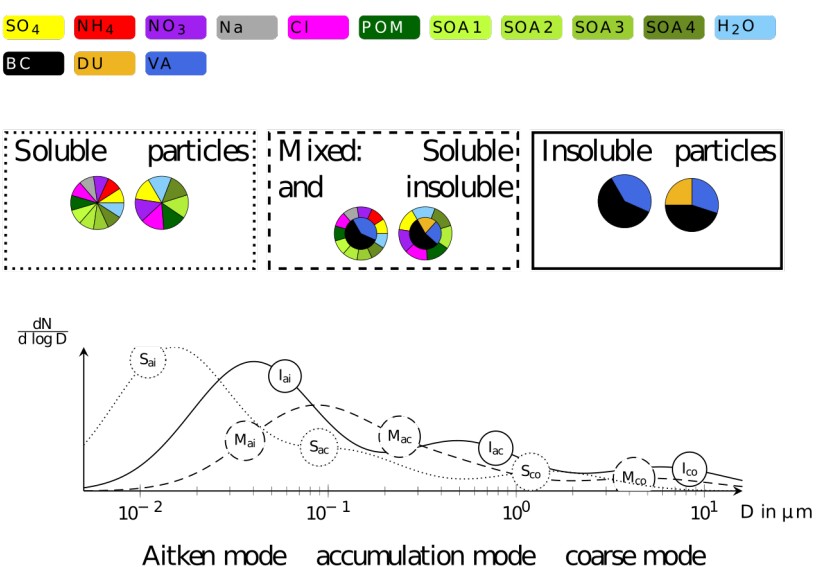

**Figure 1.** Chemical composition of the soluble (first row) and insoluble (second row) modes, mixing state of the modes (third row) and particle size distribution (giant mode is not shown). The dotted line represents a particle size distribution of soluble particles, the dashed line of mixed particles, and the solid line of insoluble particles, respectively. POM: primary organic matter, SOA: secondary organic aerosols, BC: black carbon. DU: desert dust, VA: volcanic ash. Upper panel adopted from Kaiser et al. (2014). In the current work, insoluble mode contains volcanic ash only while soluble mode contains only $SO_4^{2-}$ and $H_2O$.

$$\frac{\partial}{\partial t} M_{3,i} = -Ca_{3,ij} + Co_{3,i} + Nu_3 \tag{2}$$

where $M_{0,i}$ and $M_{3,i}$ describe the zeroth (number density) and third (mass concentration) moment of mode $i$, respectively. The terms $Ca$, $Co$ and $Nu$ refer to coagulation, condensation and nucleation, respectively. The terms $Ca_{m,ii}$ and $Ca_{m,ij}$ are intra and inter-modal coagulation in the moment $m$, respectively. Nucleation is considered for the Aitken mode only.

Condensation and coagulation affect all modes except the giant mode. The nucleation, condensation and coagulation terms are calculated following Riemer et al. (2003) and Vogel et al. (2009). Furthermore, ISORROPIA II model is used to calculate the gas-aerosol partitioning according to thermodynamic equilibrium (Fountoukis and Nenes, 2007).

Shifting between modes is performed using two mechanisms. The first mechanism is activated when a threshold diameter is exceeded. Then, a shift to a corresponding mode with larger median diameter is performed. The second mechanism shifts mass

and number concentration from insoluble modes to mixed modes if a mass threshold of soluble coating on insoluble particles (currently 5 %) is exceeded (Weingartner et al., 1997).

### 2.2.3 Aerosol optical properties

The RRTM requires the mass extinction coefficient $k_e$, single scattering albedo $\omega$, and asymmetry parameter $g$ in 30 wavelength bands to account for the radiative effect of aerosols (Gasch et al., 2017). In this connection, $k_e$ can be interpreted as the extinction cross-section per aerosol mass in the units $\mathrm{m^2\,kg^{-1}}$. The wavelength bands range between 0.2 and 100 µm. The calculation of the optical properties is based on the wavelength-dependent refractive indices of volcanic ash (Walter, 2019), water, and sulfuric acid (Gordon et al., 2017).

No study so far has treated volcanic ash as a core in an internal mixture. It is suggested, but not proven, that most volcanic ash particles are coated to some degree (Bagnato et al., 2013; Hoshyaripour et al., 2015). Therefore, the core-shell treatment is physically more realistic than the external-mixture treatment even though the reality lies between the externally mixed and core treatments (Jacobson, 2000; Riemer et al., 2019). Hence, this study deploys both externally mixed (in the soluble and insoluble modes) and internally mixed (in the mixed mode) treatments. For the mixed mode, we use the core-shell model in which the core and shell consist of well-mixed volcanic ash and $H_2O$-$H_2SO_4$ solution, respectively. To calculate the optical properties, the Mie code for coated spheres is used which has been developed by Mätzler (2002) and Bond et al. (2006) based on Bohren and Huffman (1983). Based on the ICON-ART simulations the shell fraction (increased diameter due to coating) is assumed to be 0.2 with 50 % $H_2O$-$H_2SO_4$ solution. The volume-average mixing rule is used to compute the complex refractive index of each layer, which then serves as input for the core-shell calculation.

The results of the Mie calculations for the ash-containing modes are shown in Fig. 2. It can be seen that the mixed modes (coated ash) have higher $k_e$ and $\omega$ in the visible range than the insoluble modes (uncoated ash). This is caused by the $H_2O$-$H_2SO_4$ coating which is a strong scatterer. Particles with a strongly absorbing core coated by a weak absorber generally absorb more sunlight than an external mixture of the same components, which is caused by the increase of the core cross section due to coating (Jacobson, 2000). This is not the case for volcanic ash as it is not a strong absorber compared to soot particles. This can be seen in the imaginary part of refractive indices, i.e., absorbing part, at 500 nm that are 0.00092 and 0.74 for volcanic ash and soot, respectively.

The Mie theory assumes that the particles have spherical shapes. In reality, volcanic ash particles are exclusively non-spherical particles (Bagheri and Bonadonna, 2016). Therefore, their optical properties may be better represented by spheroids, ellipsoids or even more complex structures (Gasteiger et al., 2011; Vogel et al., 2017). However, the liquid coating can lead to spherical particle surfaces, which justifies the assumption of the particle sphericity in the mixed mode. For consistency reasons, the sphericity assumption is also applied to the insoluble mode that contains uncoated ash particles. Implementing coated non-spherical ash particles into ICON-ART remains the subject of future work.

### 2.2.4 Model configuration

In the scope of this study we performed four global simulations with the ICON-ART model. The simulations run on a R3B07 grid that is also used by the German Meteorological Service (DWD) for operational weather forecasts. The horizontal grid resolution is on average $\Delta \bar{x} = 13.2$ km. 90 vertical levels resolve the atmosphere up to 75 km. The time step $\Delta t$ is 60 s. Each

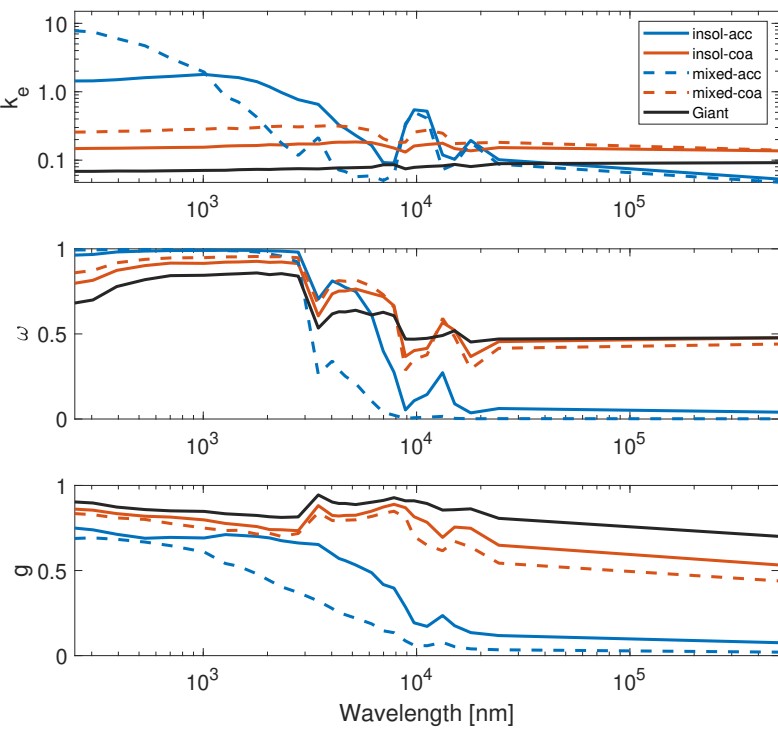

**Figure 2.** Optical properties of the ash-containing modes at RRTM wavelengths. $k_e$ has the unit $\mathrm{m^2\,g^{-1}}$. $\omega$ and $g$ are unitless. Insoluble and mixed states are shown by solid and dashed lines while accumulation and coarse sizes are demonstrated with blue and red colors, respectively.

simulation is started on 21 June 2019 at 12:00 UTC based on initialized analysis data provided by DWD. The simulation covers the first four days after the onset of the eruption.

    The volcanic emission starts on 21 June 2019, at 18:00 UTC and lasts 9 h. The simulated Raikoke eruption emits ash particles and $SO_2$. In the model the emission is characterized by an emission height and emission rate which we derived from a combined approach of satellite measurements and 1D plume simulations.

The plume height estimate is based on the MODIS and VIIRS data shown in Fig. 3. The dedicated ash algorithm (lower panel) is much more restrictive than the standard cloud-top height algorithm (upper panel), but produces similar heights where it is applied. In general, both of these brightness temperature-based products indicate maximum plume heights in the 12–12.6 km range for the time period 7–9 h after the eruption. The estimated height uncertainty is $\sim 1.5$ km. Based on this plume height estimate and also other studies (Sennet, 2019), the Raikoke eruption emits ash and $SO_2$ in our simulations at a constant

eruption rate between 8 and 14 km above sea level.

    The eruption rate of $SO_2$ is derived from measurements of the total emitted $SO_2$ mass. According to the TROPOMI (Sect. 2.1.1) and AHI data (Sect. 2.1.2), in our simulation $1.5 \times 10^9$ kg of $SO_2$ is emitted over the eruption period. To estimate the

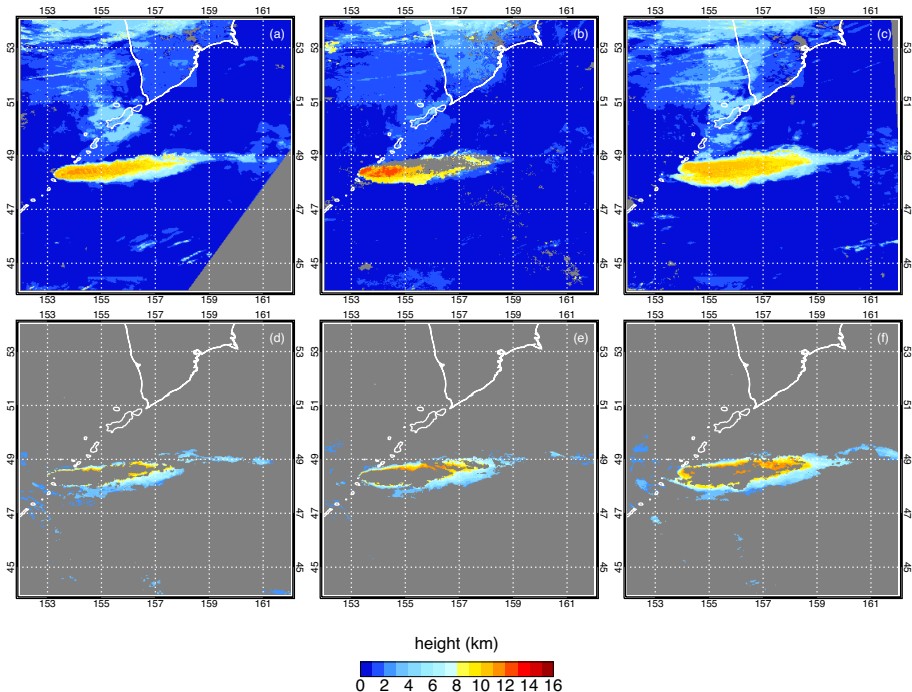

**Figure 3.** Plume height on 22 June 2019, at 01:25 UTC ((a) and (d)), 02:15 UTC ((b) and (e)), and 03:10 UTC ((c) and (f)). The top row shows standard cloud-top heights for (a) MODIS Terra, (b) VIIRS Suomi-NPP, and (c) MODIS Aqua. The bottom row plots ash heights from NOAA's dedicated volcanic ash algorithm for VIIRS on ((d) and (f)) NOAA-20 and (e) Suomi-NPP, considering only those pixels that potentially contain volcanic ash.

total mass eruption rate of volcanic ash, several 1D plume simulations using Plumeria (Mastin, 2007) and FPlume (Folch et al., 2016) are conducted assuming the following parameter ranges: plume height 12–14 km, vent diameter 90–110 m, exit velocity 100–120 m s$^{-1}$, exit temperature 900–1100 °C, and exit gas mass fraction 3 %. For this purpose, atmospheric profiles are obtained from ERA-Interim (Dee et al., 2011) and introduced in the 1D models as wind and no-wind atmospheres. By this method, the key sources of uncertainty are considered in the estimation of mass eruption rate. The results are in the range of 1.45–9.95 × 10$^6$ kg s$^{-1}$. Taking the mean value 5.7 × 10$^6$ kg s$^{-1}$ suggests that about 190 × 10$^9$ kg tephra is emitted within 9 hours. Assuming that 1 % of the erupted mass is very fine ash with $d < 30$ μm (relevant for long range transport) (Rose and Durant, 2009; Gouhier et al., 2019), we estimate that 1.9 × 10$^9$ kg very fine ash is injected into the atmosphere during the eruption. The estimates by the 1D models are in agreement with AHI data (Sect. 2.1.2).

The estimated 1.9 × 10$^9$ kg of very fine ash are used in the ICON-ART simulations and distributed equally between accumulation, coarse, and giant modes. The number concentration of the log–normal distribution is calculated based on the median diameter $d_e$ and standard deviation $\sigma_e$ of the emitted particle distribution. Table 1 lists details about these emitted particle size distributions. They are based on data from Bonadonna and Scollo (2013).

**Table 1.** Emission parameters for ash emission with median diameter $d_e$ and standard deviation $\sigma_e$ of ash size distribution, and the mass emission rate $Q_e$ of each ash mode and $SO_2$.

| Ash mode | Accumulation | Coarse | Giant | $SO_2$ |
|---|---|---|---|---|
| $d_e$ [μm] | 0.8 | 2.98 | 11.35 | – |
| $\sigma_e$ [-] | 1.4 | 1.4 | 1.4 | – |
| $Q_e$ [$\mathrm{kg\,s^{-1}\,m^{-1}}$] | 3.26 | 3.26 | 3.26 | 7.72 |

We study the effect of aerosol dynamic processes and the radiative effect of internally mixed particles on the volcanic plume dispersion with the help of four different simulation scenarios summarized in Table 2. The first scenario (AERODYN-rad) uses the whole new development of the AERODYN module together with the radiative feedback of internally mixed particles. In the second scenario (no_AERODYN-rad) only insoluble ash particles of three different size ranges are transported. Secondary aerosol formation and particle aging are switched off. However, the volcanic ash still interacts with solar and thermal radiation. The third scenario (AERODYN-no_rad) considers the effects of aerosol aging without any radiative feedback of these particles. The fourth scenario represents the status quo of operational volcanic cloud forecast. It considers neither aerosol dynamic effects nor aerosol-radiation interaction.

The two scenarios with AERODYN treat $SO_2$ as a chemical substance which can be oxidized. The chemical reaction scheme is a simplified OH-chemistry scheme that has been implemented into ICON-ART by Weimer et al. (2017). The no_AERODYN scenarios treat $SO_2$ as a passive tracer without any gas phase chemistry.

**Table 2.** Simulation scenarios with their represented processes.

| scenario | aerosol dynamics and gas phase chemistry | aerosol–radiation interaction |
|---|---|---|
| AERODYN-rad | on | on |
| no_AERODYN-rad | off | on |
| AERODYN-no_rad | on | off |
| no_AERODYN-no_rad | off | off |

## 3  Results and Discussion

### 3.1  Ash and $SO_2$ transport

We compare our model results with different satellite products as introduced in Sect. 2.1. Figure 4 (a) and (b) show daily mean AHI retrievals of volcanic ash mass loading. As described earlier, the filtered data is used. For the daily mean only ash containing pixels are considered. The same averaging approach we apply on the ICON-ART model results, shown in panels (c) to (f) of Fig. 4. Panels in the left column show measurements and model results of 22 June 2019, panels in the right column

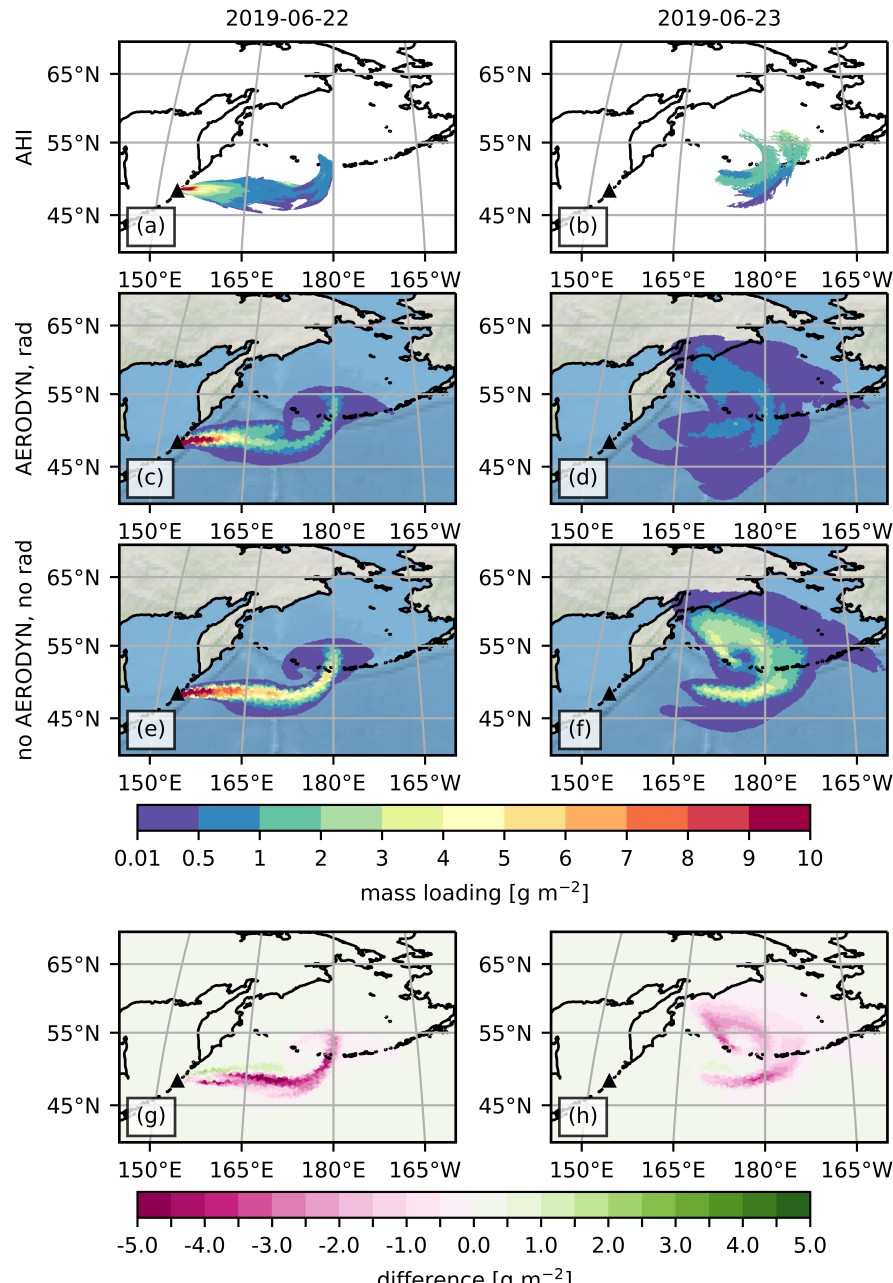

**Figure 4.** Daily mean total column mass loading of volcanic ash on 22 June (left column) and 23 June 2019 (right column). Top row (panel (a) and (b)) shows results measured by AHI on-board Himawari-8. The middle and lower row (panel (c) – (f)) show ICON-ART results for AERODYN-rad and no_AERODYN-no_rad, respectively. The black triangle depicts the location of Raikoke volcano. Panels (g) and (h) show the absolute difference between the two simulation scenarios.

of 23 June, respectively. On 22 June the volcanic cloud moved eastward towards $180°$ E where the direction of transport turned northward. The maximum of daily mean mass loading is still located in proximity to the volcano. For this day, both model results and the satellite retrieval agree very well in location, structure, and absolute values of ash mass loading. We can assume that the model captures the atmospheric state well, one day after its initialization. Furthermore, there are only minor differences between the two different simulation setups for the results of 22 June in Fig. 4 (c) and (e). These differences are mainly restricted to the slightly higher mass loading in panel (e) and small differences in the volcanic cloud structure. For the first day after the eruption, the aerosol dynamic effects and the aerosol–radiation interaction have only a minor influence on the volcanic ash mass loading. On 23 June the averaged AHI measurements show a more fragmentary ash distribution in Fig. 4 (b). This might be a result of volcanic cloud dilution in combination with deficiencies in the volcanic ash measurement of opaque regions. Most of the ash is measured between $50–55°$ N and around $180°$ E. The simulation results in Fig. 4 (d) and (e) support the assumption of the diluted volcanic cloud, as the mass loading only shows values smaller than $4\,\mathrm{g\,m^{-2}}$. For both simulated scenarios, the overall structure of the volcanic cloud is similar. However, differences prevail in location and absolute values of maximum mass loading. These differences are due to aerosol dynamics and radiative effects which are addressed in more detail in Sect. 3.2 and Sect. 3.3, respectively. Compared to these two simulations, the averaged AHI measurements (Fig. 4 (b)) show values for the maximum ash mass loading that lie in between the two simulation scenarios. In panels (g) and (h) the differences between the two are highlighted by the absolute difference of AERODYN-rad – no_AERODYN-no_rad. It shows that considering aerosol dynamics and aerosol-radiation interaction results in lower volcanic ash mass loadings in most parts of the volcanic cloud. Only for the first day after the eruption, the volcanic cloud seems to be shifted slightly north in the AERODYN-rad scenario compared to the no_AERODYN-no_rad scenario, as the difference plot shows some positive values between $160–170°$ N.

In order to compare our ICON-ART results in an objective manner with the AHI observations, we make use of the SAL method. This quality measure has been introduced by Wernli et al. (2008) and has been extensively discussed by Wernli et al. (2009). The method identifies objects in a 2D field (e.g., total ash mass loading) and quantifies the differences between model and observation in structure (S), amplitude (A), and location (L). A value of $0$ implies perfect agreement. We apply the SAL method with a fix threshold value to identify objects $R^* = 0.01\,\mathrm{g\,m^{-2}}$. The results for the comparison of daily mean total column mass loading between the AHI retrieval and the ICON-ART results are summarized in Table 3. The location of the

**Table 3.** Comparison of daily mean total column mass loading of volcanic ash between AHI and ICON-ART results using the SAL method by Wernli et al. (2008).

| scenario | 2019-06-22 | | | 2019-06-23 | | |
|---|---|---|---|---|---|---|
| | S | A | L | S | A | L |
| AERODYN-rad | −0.191 | 0.584 | 0.004 | 1.651 | 0.298 | 0.041 |
| AERODYN-no_rad | −0.323 | 0.579 | 0.002 | 1.362 | 0.275 | 0.028 |
| no_AERODYN-rad | −0.202 | 0.921 | 0.014 | 1.601 | 0.716 | 0.031 |
| no_AERODYN-no_rad | −0.270 | 0.874 | 0.013 | 1.546 | 0.748 | 0.030 |

volcanic cloud agrees very well with the observation for all dates in all simulation scenarios. The structure of the volcanic
cloud shows larger differences compared to observations, especially on 23 June. However, the values are rather similar for
the different simulation scenarios. Only the amplitude values differ distinctly among the different scenarios. Simulations with
AERODYN are closer to the observation than simulations without aerosol dynamics.

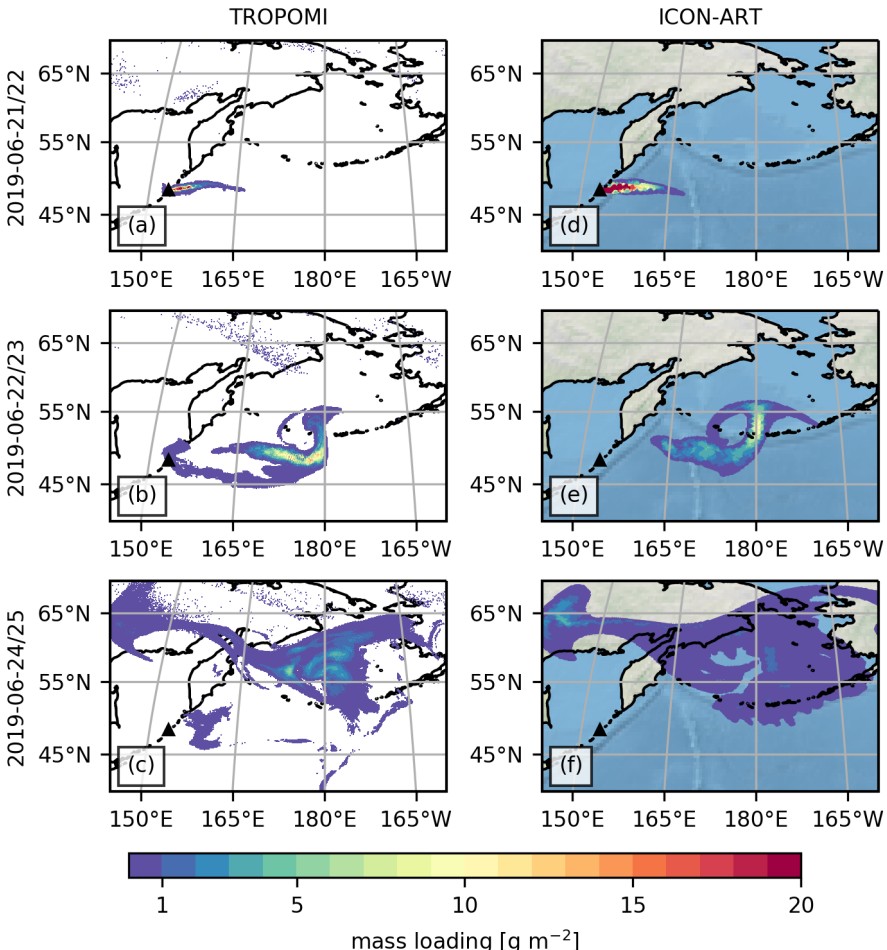

**Figure 5.** Mass loading of SO$_2$ measured by TROPOMI during three different time periods are shown in panels (a), (b), and (c). Panels (d), (e), and (f) show ICON-ART results of AERODYN-rad at corresponding time steps.

Figure 5 shows three TROPOMI retrievals of SO$_2$ mass loading in $\mathrm{g\,m^{-2}}$ in panels (a), (b), and (c) for three different dates. Each of these three graphs is a composite of several satellite orbits, chosen from a batch of 14 consecutive orbits (approximately 24 h coverage). Those orbits that directly detect the volcanic cloud in Fig. 5 (a) intersected with the area of interest (see Sect. 2.1.1) on 22 June 2019, between 02:16 and 02:29 UTC. Data points containing the volcanic cloud signature in Fig. 5 (b) were measured on 23 June, between 00:15 and 02:10 UTC and in Fig. 5 (c) between 24 June, 20:16 UTC and 25 June, 03:13 UTC,

respectively. Panels (d) to (f) show ICON-ART results of AERODYN-rad for three different time steps. These time steps have been chosen to be closest to the mean of the time period of the corresponding TROPOMI measurement. The overall structure

of the $SO_2$ mass loading agrees well between model results and observations. This is especially true for the two earlier dates when the modelled atmospheric state can be assumed to be closer to reality than for later dates. But also the model result $3.5$ days after its initialization in Fig. 5 (f) shows very good agreement with the TROPOMI measurement in (c). A main difference between satellite retrieval and model result is the location of the maximum $SO_2$ mass loading. Although the magnitude of the maximum $SO_2$ mass loading is in good agreement, in the model results its location appears further downstream compared

to the satellite measurement. One reason could be the different time of measurement and model result. However, a greater influence can be expected by uncertainties of the emission profile parametrization and of the simulated wind velocities. In case more $SO_2$ is emitted in altitudes with higher wind speeds in the model, it will be transported faster. The same applies for the case that in some altitudes wind speeds in the model are slightly higher than they are in reality. Furthermore, the TROPOMI measurements can also be erroneous. The TROPOMI sensor might not capture all of the $SO_2$ due to deficiencies of

the measurement technique in opaque regions. Assumptions about a vertical $SO_2$ profile made for the retrieval can also result in incorrect $SO_2$ mass loadings.

    The AHI and TROPOMI measurements give us confidence in the simulated horizontal distribution of the volcanic cloud. Additionally, we retrieve information about the vertical extension of the volcanic cloud from OMPS-LP and CALIOP data. OMPS-LP gives a clear signal of the volcanic cloud on 22 June 2019, 02:27 UTC shortly after the onset of the eruption. It

locates the volcanic cloud at $49.76°$ N $154.1°$ E at approximately $17\,\mathrm{km}$. The ICON-ART model result (AERODYN-rad) shows a similar cloud top height which will be addressed in more detail in Sect. 3.3. Also the height of the volcanic cloud measured by CALIOP on 23 June 2019, agrees well with the model result. This will be addressed in more detail in the following section.

### 3.2   Effect of aerosol dynamics

So far we mainly discussed the ICON-ART model result of the AERODYN-rad scenario. In this section, we compare it with

the no_AERODYN-rad scenario to study the influence of secondary aerosol formation and particle aging on volcanic aerosol dispersion.

    The CALIPSO satellite passed over the volcanic cloud on 23 June 2019, at around 15:00 UTC. On this date, the satellite ground track clearly intersects the modeled volcanic cloud, as shown in Fig. 6 (a). The 2D map depicts the volcanic cloud top height of accumulation mode ash particles calculated with ICON-ART (AERODYN-rad). In this connection, a threshold of

$0.01\,\mathrm{\mu g}$ ash per $\mathrm{kg}$ air defines the volcanic cloud top. The map shows a maximum volcanic cloud top height in the range of $17\text{–}19\,\mathrm{km}$ under the CALIPSO ground track at around $50°$ N. The CALIOP measurement for the total attenuated backscatter at $532\,\mathrm{nm}$, shown in Fig. 6 (b), indicates volcanic aerosols between $49°$ N and $51°$ N at height levels between $15$ and $16\,\mathrm{km}$. Attenuated backscatter at $532\,\mathrm{nm}$ of volcanic aerosols on 23 June for the 15:00 UTC model output (AERODYN-rad) is displayed in Fig. 6 (c). Based on the simulated ash and sulfate concentrations as well as their optical properties the attenuated

backscatter is determined for model columns along the CALIPSO ground track.

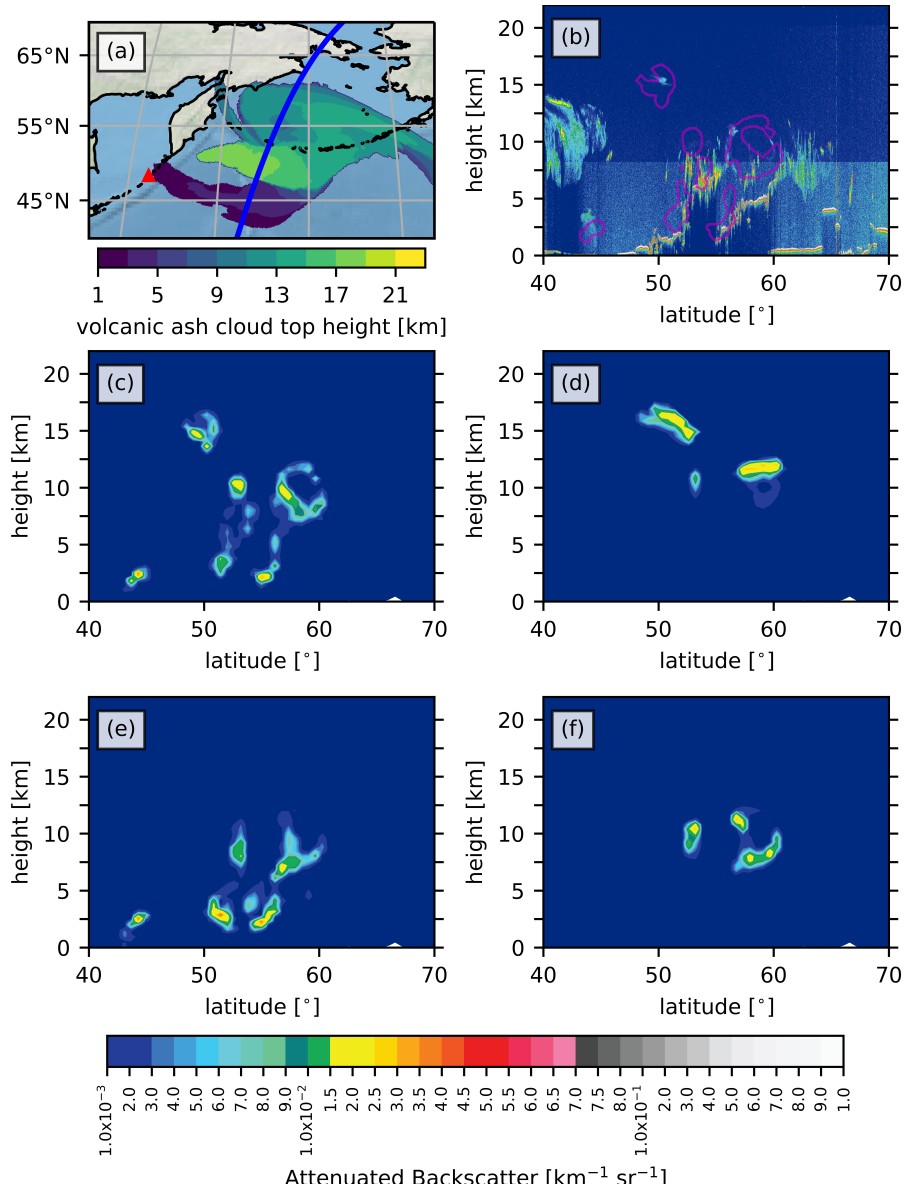

**Figure 6.** (a) CALIPSO ground track on 23 June 2019, around 15:00 UTC in blue color and location of Raikoke volcano as red triangle. The contour map shows the volcanic ash cloud top height for the AERODYN-rad scenario. (b) The CALIOP attenuated backscatter for 532 nm for the satellite position between $40°$ N and $70°$ N is displayed in the top right panel. The magenta line shows the $0.002\,\mathrm{km^{-1}\,sr^{-1}}$ contour of AERODYN-rad at 15:00 UTC. Middle and lower panels: Total attenuated backscatter for 532 nm of volcanic aerosols under the CALIPSO ground track on 23 June 2019, for the 15:00 UTC model output are displayed. (c) shows the result for AERODYN-rad, (d) for no_AERODYN-rad, (e) for AERODYN-no_rad, and (f) for no_AERODYN-no_rad, respectively.

Our model result (AERODYN-rad) captures the most prominent feature of the CALIOP retrieval between $49°$ N and $51°$ N at a height around $16$ km. Here, the model shows a clear maximum in total attenuated backscatter of volcanic aerosol. Furthermore, the model result shows several other peaks in attenuated backscatter. In order to make the model result in panel (c) better comparable with the measurement, the magenta line in panel (b) shows the $0.002\,\mathrm{km}^{-1}\,\mathrm{sr}^{-1}$ contour of the model result.

For example, the peak in the simulated attenuated backscatter (Fig. 6 (c)) at around $44°$ N up to 3 km is also present in the CALIOP signal at a comparable order of magnitude. This suggests that the elevated CALIOP signal in this region is due to volcanic aerosols. Other features in the modeled attenuated backscatter, north of $51°$ N, also collocate with structures in the CALIOP signal. This suggests that part of the elevated CALIOP signal in these regions is due to the volcanic aerosol cloud. It nicely shows the advantage of considering model results for the interpretation of satellite retrievals.

Comparing AERODYN-rad in Fig. 6 (c) with no_AERODYN-rad in Fig. 6 (d) shows the distinct effect of aerosol dynamics on vertical distribution of the volcanic cloud. No_AERODYN-rad catches the main feature between $49°$ N and $51°$ N at a height up to $17$ km. However, the volcanic aerosol layer extends significantly further north, up to $54°$ N. This is in contrast to the CALIOP signal in Fig. 6 (b). Also the smaller patterns in lower altitudes and higher latitudes are missing in the no_AERODYN-rad scenario. The same applies for the feature at around $44°$ N and 3 km height. Without aerosol dynamics, most of the aerosol

stays at one height level, whereas with aerosol dynamics, the particles get also mixed down to lower altitudes. Coagulation of particles and condensation of sulfate and water onto existing particles increases the aerosol mass. Hence, these particles sediment faster and therefore, are removed from the atmosphere more efficiently.

A similar conclusion can be derived from the AERODYN-no_rad and no_AERODYN-no_rad scenarios in Fig. 6 (e) and (f), respectively. Although, both are missing the most prominent feature between $49°$ N and $51°$ N at around $16$ km, they show the

same behavior in terms of aerosol dynamic effects.

Additional dates of CALIPSO measurements are displayed in Appendix A.

To further investigate the effect of aerosol dynamics on the residence time of very fine ash, we examine the temporal variation of ash concentration in the atmosphere. This is illustrated in Fig. 7. The graph shows how the normalized total ash mass $\widetilde{m}_{ash}$ evolves over time after the onset of the volcanic eruption on 21 June 2019, at 18:00 UTC. We define

$$\widetilde{m}_{ash}(t) = \frac{m_{ash}(t)}{\max(m_{ash}(t))}$$

with $m_{ash}(t)$ as the total observed volcanic ash mass at one measurement time or simulation time step, respectively. In the ICON-ART simulations, AERODYN-rad and no_AERODYN-rad, $\max(m_{ash}(t))$ is close to $1.9 \times 10^9$ kg. For the AHI retrieval $\max(m_{ash}(t))$ is estimated to range between $0.4 \times 10^9$ and $1.8 \times 10^9$ kg. Figure 7 shows $\widetilde{m}_{ash}$ for two different simulation scenarios, AERODYN-rad (green) and no_AERODYN-rad (yellow), and the AHI retrieval (black). The gray shading depicts

an error estimate for the AHI measurement between $0.4\widetilde{m}_{ash}$ and $1.6\widetilde{m}_{ash}$.

Both simulations and the satellite measurement agree very well over the course of the first 9 h. This is the eruption phase of the Raikoke volcano. As Raikoke did not erupt continuously over these 9 h, the offset between simulation and observation as well as the small-scale variations in the observation during this period can be explained. The main more or less continuous

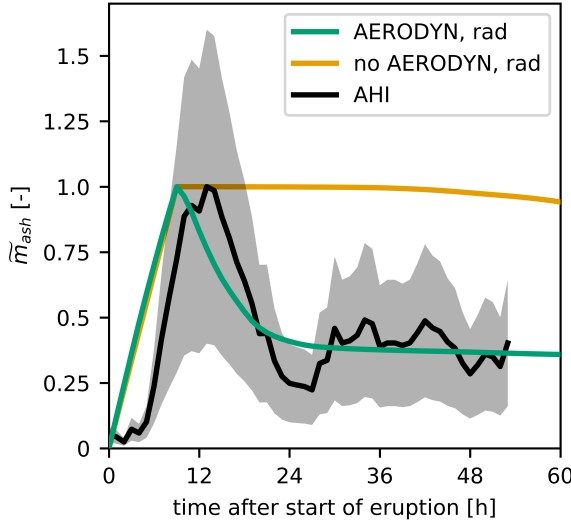

**Figure 7.** Normalized total volcanic ash mass $\widetilde{m}_{ash}$ over the time after the onset of the volcanic eruption on 21 June 2019, at 18:00 UTC. The green and yellow curve represent AERODYN-rad and no_AERODYN-rad, respectively. The black curve is based on AHI measurements with an error estimate in gray.

eruption of Raikoke occurred between 21 June 2019, 22:40 UTC and 22 June, 02:00 UTC; with several additional puffs before
and after this period. While in the model we assumed a constant and continuous eruption.

After the end of the eruption, the observed ash mass (black) decays to less than 50 % over the course of 12 h. Thereafter, the total volcanic ash mass seems to stabilize. The small-scale variations in the observation might be due to deficiencies or limitations of the retrieval algorithm, as no new ash is emitted during this period. We can see a very similar decay and stabilization of ash mass for the AERODYN-rad scenario in green. The result suggests that the necessary sink processes are represented by
our new aerosol dynamics module. The same are missing in no_AERODYN-rad, for which the volcanic ash mass decays much slower. We deduce that secondary aerosol formation and particle aging, due to condensation and coagulation, are essential processes for the correct simulation of volcanic aerosol dispersion. These processes largely influence the transported aerosol concentrations. Additionally, we would like to note that the prevailing settling mechanism of aerosol after the Raikoke 2019 eruption for all our simulation scenarios is due to sedimentation. Dry deposition is only relevant for aerosol near the ground.
Wet deposition should also play a minor role during the first days after the eruption, as most of the volcanic ash is emitted above cloud level.

### 3.3 Effect of radiative interaction

In contrast to aerosol dynamics, aerosol–radiation interaction does not largely influence the transported aerosol concentrations. This can be deduced from the SAL analysis in Table 3. There are only minor differences in the amplitude of volcanic ash mass

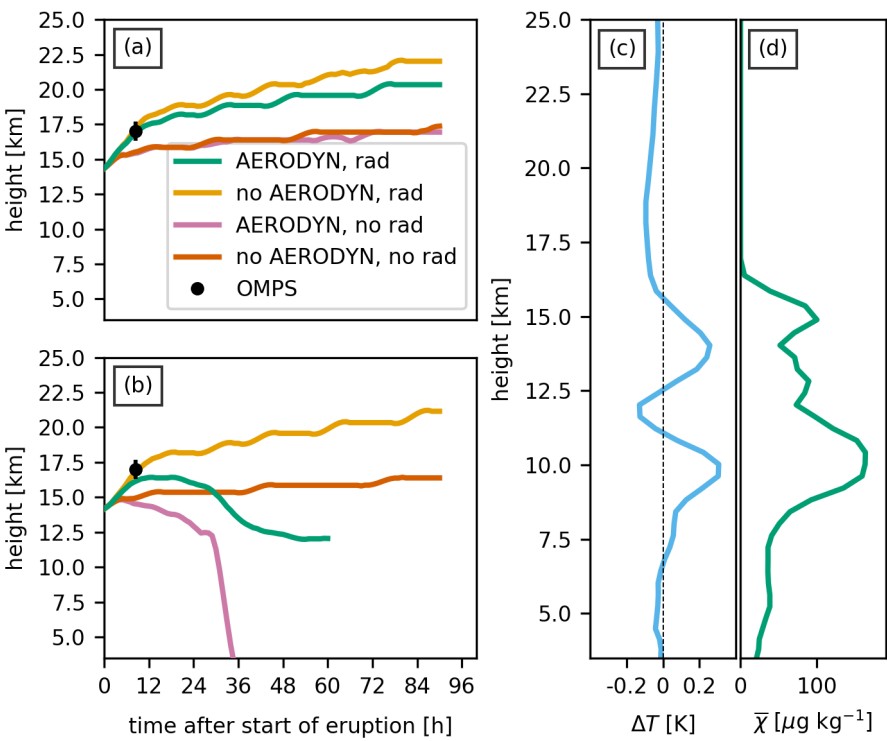

**Figure 8.** (a) and (b) Evolution of height of volcanic ash cloud top after the onset of the eruption on 21 June 2019, at 18:00 UTC. The yellow curve represents the no_AERODYN-rad scenario, the green curve AERODYN-rad, the pink one AERODYN-no_rad, and the orange one represents the no_AERODYN-no_rad scenario. Panel (a) shows the ash cloud top of particles in the accumulation mode, (b) of particles in the coarse mode, respectively. The black circle depicts the volcanic cloud top height obtained from OMPS-LP. (c) Mean temperature difference (AERODYN-rad – AERODYN-no_rad) in volcanic ash cloud columns on 23 June 2019, 12:00 UTC. (d) Mean volcanic ash concentration $\overline{\chi}$ for the same model columns as in (c) for AERODYN-rad.

loading comparison between the two scenarios with or without radiation-interaction, but the same aerosol dynamics setup. However, there are differences in the mass loading patterns that can be explained by radiative effects. This is already somewhat indicated by the S value in Table 3. The S values of simulation scenarios with the same radiation-interaction setup are closer to each other compared to the other scenarios.

In order to investigate the influence of aerosol–radiation interaction on volcanic plume dispersion in more detail, we look at

the maximum height that the volcanic cloud reaches over the course of time. A volcanic cloud that is lifted up in the atmosphere has a longer lifetime. Hence, it can be transported over longer distances, remains a hazard for aircraft over a longer period of time, and has longer lasting climatic effects. Additionally, the height of the volcanic cloud in the atmosphere also influences its transport, as wind speed and direction can differ between height levels. Figure 8 (a) and (b) show the height of the volcanic cloud top over the course of time after the onset of the volcanic eruption. We used a threshold value to determine the extent

of the volcanic cloud in the model result. A model grid box with an ash concentration above this threshold is considered as

part of the volcanic cloud. For accumulation mode ash particles this threshold is set to $0.01\,\mu g\,kg^{-1}$ and for coarse mode ash particles to $1\,\mu g\,kg^{-1}$. The different colours in Fig. 8 (a) and (b) represent the four different simulated scenarios. The upper panel shows the volcanic cloud top height of ash particles in the accumulation mode. The lower one shows the same graph for ash particles in the coarse mode.

Comparing the yellow (no_AERODYN-rad) with the green curve (AERODYN-rad), we can see the influence of the aerosol dynamic processes on the maximum volcanic cloud top height. For both, the accumulation and the coarse mode the volcanic cloud top height is lower for the scenario with AERODYN. This result agrees with the backscatter signal of the same two simulation scenarios in Fig. 6. Due to aerosol dynamic processes particles grow in size as they age over time. Hence, the volcanic cloud is located at lower altitudes. This effect is more pronounced for the larger and therefore heavier coarse mode

particles. Due to their larger surface, the condensation of sulfate onto them is more efficient compared to accumulation mode particles. The result indicates that for coarse mode ash the aging process is the determining factor of whether the volcanic cloud rises higher or sinks. The ash cloud top height of coarse mode ash particles in no_AERODYN-rad continuously rises up to more than $20\,km$. In contrast, the ash cloud top height in AERODYN-rad gradually sinks during the following $50\,h$ (after reaching its peak). The graph for the AERODYN-rad scenario stops after around $60\,h$. This behaviour can be explained by

the evaluation method. The aged coarse mode particles sediment out and reduce their concentration significantly. Eventually, the concentration sinks to the same order of the threshold value that is used to determine the volcanic cloud. From this point onward, the maximum volcanic cloud top height cannot be determined reliably anymore.

Even more pronounced than the aerosol dynamic effect, we can see the influence of radiative effects on the volcanic cloud dispersion in Fig. 8. A distinct difference prevails between the two scenarios with radiative interaction (yellow and green curve)

and the two without radiative interaction (pink and orange curve). Accumulation mode ash particles stay more or less at the initial maximum height level ($14\,km$) in case they do not interact with radiation. On the contrary, the ash cloud top rises up to $20\,km$ in the two scenarios with radiative interaction over the first four days after the onset of the eruption. Furthermore, the graph for accumulation mode ash particles indicates that the aerosol aging reduces the lifting effect induced by radiative interaction by higher sedimentation velocities due to larger particles. Hence, pure ash particles are lifted higher compared to

aged ash particles.

The described behavior is even more pronounced for coarse mode ash particles, shown in Fig. 8 (b). Especially for the simulated scenario with no radiative interaction, but aerosol dynamic processes (pink curve), the ash particles sediment out over the course of the first $30\,h$ after the onset of the eruption. In contrast, the two scenarios with radiative interaction again show a lifting in volcanic cloud top height over the first $12\,h$. Subsequently, the influence of particle aging becomes more

relevant for coarse mode ash particles. As for accumulation mode particles, in the no_AERODYN-no_rad scenario (orange curve) coarse mode particles also tend to stay on the same height level.

A direct effect of the radiative interaction is shown in Fig. 8 (c) and (d) exemplarily for the model result of 23 June 2019, 12:00 UTC. The graph in (c) depicts the horizontally averaged atmospheric temperature difference $\Delta T$ between AERODYN-rad and AERODYN-no_rad at different heights. For the averaging approach, only model columns which contain a volcanic

ash mass loading $> 0.01\,g\,m^{-2}$ in both scenarios are considered. Figure 8 (d) illustrates the horizontally averaged volcanic

ash concentration $\overline{\chi}$ at different heights for the AERODYN-rad scenario. For this averaging we consider exactly the same model columns as we use for the temperature difference. The curve of the temperature difference shows two distinct peaks, one at around 10 the other at around $14\,\mathrm{km}$. Here, the simulation which considers aerosol–radiation interaction exhibits around $0.25\,\mathrm{K}$ higher air temperature. Both peaks collocate with the lower and upper boundary of the volcanic ash cloud, respectively.

In these two height layers, the volcanic ash leads to an increased absorption of solar and thermal radiation, hence, it heats the surrounding air. The resulting vertical velocity perturbation $\Delta w$ is in the order of $0.1\,\mathrm{ms^{-1}}$. For this purpose, we analyzed the difference in vertical velocity between the AERODYN-rad and AERODYN-no_rad scenario during the first $12\,\mathrm{h}$ after the eruption. Only grid cells in model columns which contain a volcanic ash mass loading $> 0.01\,\mathrm{g\,m^{-2}}$ in both scenarios are considered. Locally, $\Delta w$ reaches $0.19\,\mathrm{ms^{-1}}$ with a 98th percentile of $0.05\,\mathrm{ms^{-1}}$. This agrees well with the vertical lifting of

the volcanic cloud top height of around $3\,\mathrm{km}$ during the first $12\,\mathrm{h}$ ($\overline{w} = 0.07\,\mathrm{ms^{-1}}$).

     The comparison of the four simulated scenarios with the OMPS-LP retrieval indicates that considering aerosol radiative effects is essential to simulate volcanic aerosol dispersion correctly, already over the course of the first four days after the start of the eruption. Especially the simulated height of the accumulation mode particle's cloud top in Fig. 8 (a) agrees very well with the measured height. It should be noted that the OMPS-LP measurement gives the volcanic cloud height at one

(horizontal) position. The maximum volcanic cloud top height is not necessarily collocated with this measurement position. However, at this early stage during the eruption phase the volcanic cloud is not distributed over a large area yet. That is why we assume that the volcanic cloud top height does not differ significantly in horizontal direction. Additionally, the ICON-ART model result shows the maximum volcanic cloud top height in proximity to the location of the satellite measurement. Based on the simulation result, we assume that mainly accumulation mode particles are present at the top of the volcanic cloud. These

particles are in the size of $0.1\,\mathrm{\mu m}$.

## 4   Conclusions

In the scope of this work, we use the Raikoke eruption of June 2019 as a natural experiment to investigate the influence of particle aging and aerosol–radiation interaction on volcanic aerosol dispersion. We simulate volcanic aerosol dispersion with the ICON-ART modelling system together with the newly implemented AERODYN module. The results presented allow us to

answer the posed research questions:

     1) Particle aging generates internally mixed aerosols due to condensation and coagulation. These processes generally increase particle sizes and consequently, the sedimentation velocity. Therefore, ash aging mainly influences the sink processes. As a consequence of the higher sedimentation velocity, also the vertical distribution of volcanic aerosols is affected. Our results suggest that aerosol dynamic effects lead to a removal of around $50\,\%$ of volcanic ash mass (very fine ash) over the course of

$12\,\mathrm{h}$ after the end of the Raikoke eruption on 22 June 2019.

     2) The aerosol–radiation interaction has a significant impact on the volcanic aerosol dispersion already during the very first days after the eruption. Without this interaction volcanic ash sediments out fast and does not reach height levels measured by

satellite instruments, such as OMPS-LP. Our results suggest that the Raikoke volcanic cloud top rises around 3 km during the first 12 h and reaches a height of more than 20 km after 4 days.

3) The comparison between model results and satellite retrievals, such as CALIOP and AHI, suggests that aerosol dynamic processes are crucial for the correct simulation of volcanic aerosol dispersion during the first couple of days after the eruption. Both, the aging process and the aerosol–radiation interaction influence the vertical distribution of aerosols and therefore, determine at which altitude the particles are transported. The radiative effect is responsible for the rise of the volcanic cloud top, whereas the particle aging is responsible for an efficient mixing of aerosols into lower altitudes. Furthermore, this study illus-

trates that representing sink processes correctly is necessary for the correct and reliable forecast of volcanic aerosol dispersion.

*Code and data availability.* The output from ICON-ART simulations performed in this study can be provided upon request by the corresponding author. The ICON-ART code is licence protected and can be accessed by request to the corresponding author. The NOAA Ash Height Product (Pavolonis, Michael, Qi, Hongming, and NOAA JPSS Program Office (2017): NOAA JPSS Visible Infrared Imaging Radiometer Suite (VIIRS) Volcanic Ash Detection and Height Environmental Data Record (EDR) from NDE. NOAA National Centers for Environmental Information. doi:10.7289/V5BK19KS. [Accessed in April 2020]) is available from the NOAA Comprehensive Large Array-data Stewardship System (CLASS) archive (http://www.class.noaa.gov/saa/products/search?datatype_family=JPSS_GRAN). The MODIS Cloud Product (Platnick, S., S. Ackerman, M. King, G. Wind, K. Meyer, P. Menzel, R. Frey, R. Holz, B. Baum, and P. Yang, 2017. MODIS atmosphere L2 cloud product (06_L2), NASA MODIS Adaptive Processing System, Goddard Space Flight Center, [doi:10.5067/MODIS/MOD06_L2.061; doi:10.5067/MODIS/MYD06_L2.061]) and the SNPP VIIRS Cloud Properties product (doi: 10.5067/VIIRS/CLDPROP_L2_VIIRS_SNPP.011) are available from the NASA LAADS DAAC (https://ladsweb.modaps.eosdis.nasa.gov). TROPOMI data is publicly available on https://s5phub.copernicus.eu. Himawari-8 AHI datasets that have been analyzed in the scope of this study can be provided upon request by the corresponding author. OMPS data is available after registration at https://www.iup.uni-bremen.de/DataRequest/. CALIPSO data can be found on https://eosweb.larc.nasa.gov/project/calipso/calipso_table/.

## Appendix A: Total Attenuated Backscatter at 532 nm

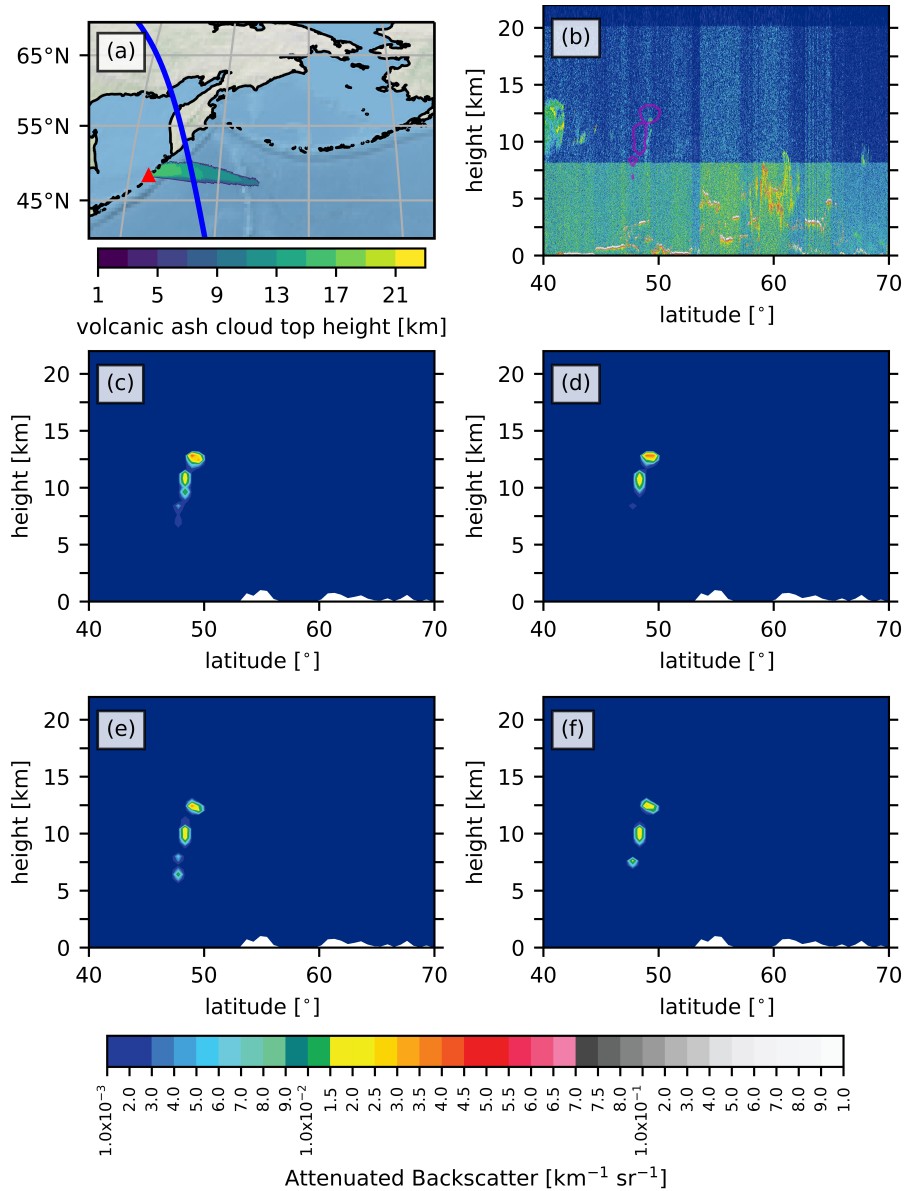

**Figure A1.** (a) CALIPSO ground track on 22 June 2019, around 03:00 UTC in blue color and location of Raikoke volcano as red triangle. The contour map shows the volcanic ash cloud top height for the AERODYN-rad scenario. (b) The CALIOP attenuated backscatter for $532\,\mathrm{nm}$ for the satellite position between $40°\,\mathrm{N}$ and $70°\,\mathrm{N}$ is displayed in the top right panel. The magenta line shows the $0.002\,\mathrm{km}^{-1}\,\mathrm{sr}^{-1}$ contour of AERODYN-rad at 03:00 UTC. Middle and lower panels: Total attenuated backscatter for $532\,\mathrm{nm}$ of volcanic aerosols under the CALIPSO ground track on 22 June 2019, for the 03:00 UTC model output are displayed. (c) shows the result for AERODYN-rad, (d) for no_AERODYN-rad, (e) for AERODYN-no_rad, and (f) for no_AERODYN-no_rad, respectively.

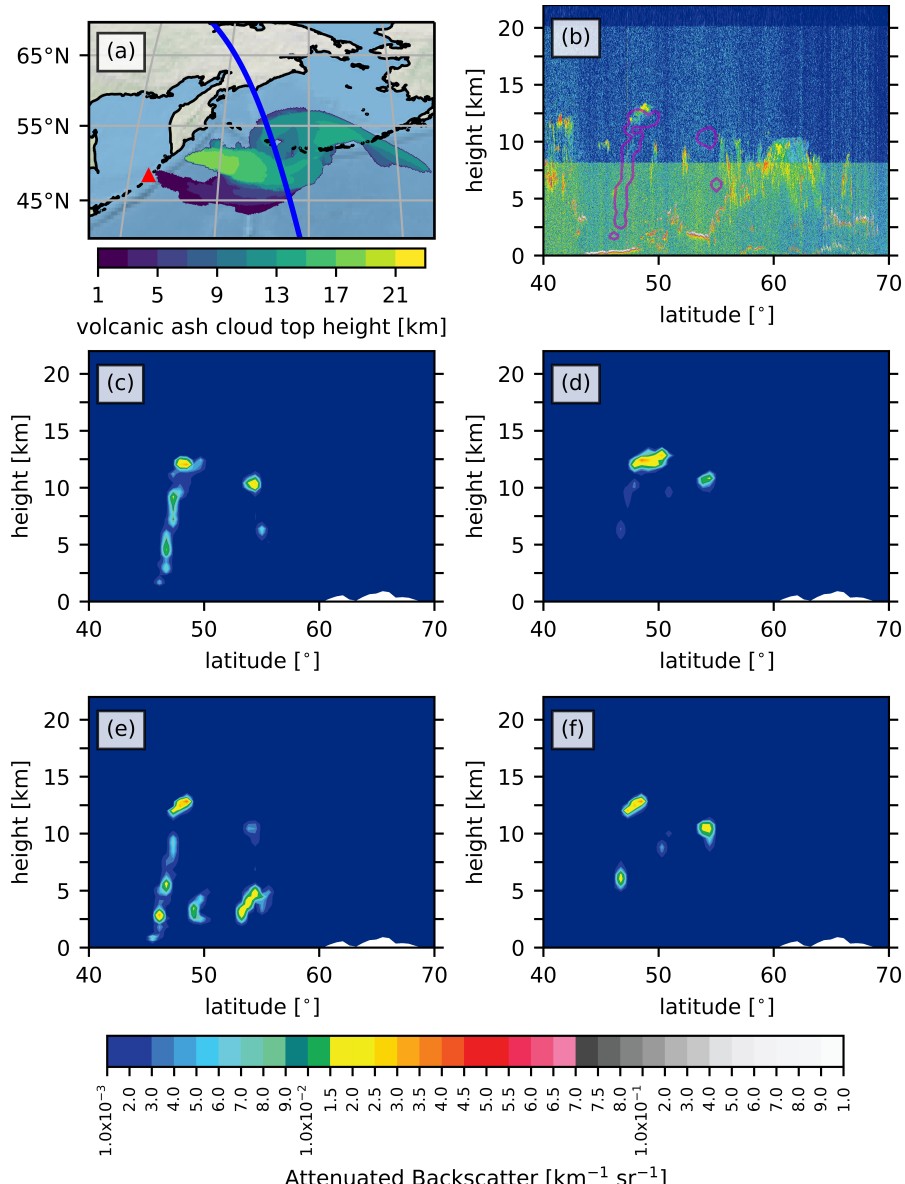

**Figure A2.** Same as Fig. A1 on 23 June 2019, 02:00 UTC.

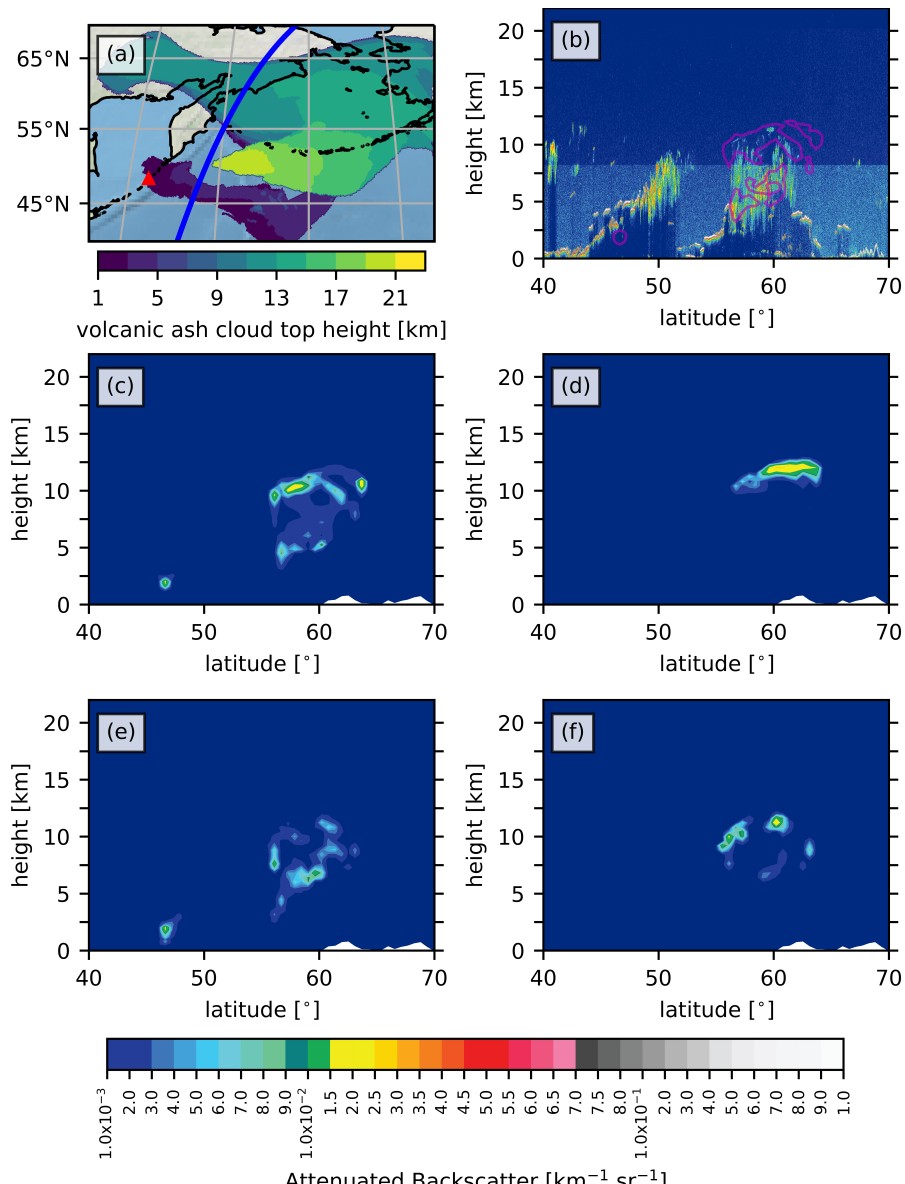

**Figure A3.** Same as Fig. A1 on 24 June 2019, 16:00 UTC.

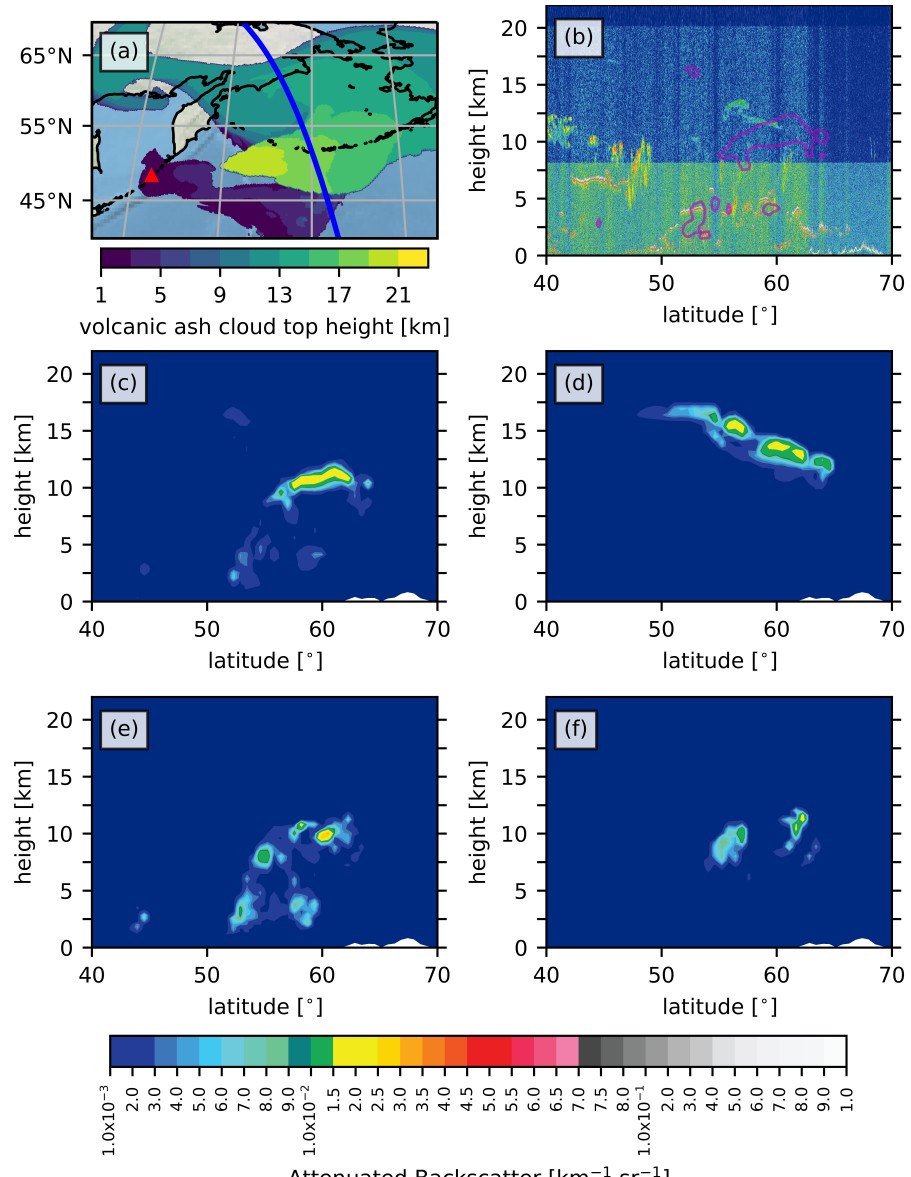

**Figure A4.** Same as Fig. A1 on 25 June 2019, 01:00 UTC.

*Author contributions.* LOM, GAH, HV, JB and BV developed the ICON-ART AERODYN code and carried out the simulations. GAH and JB conducted and analyzed 1D plume simulations. AH provided the plume height estimates based on MODIS and VIIRS data. SW contributed the TROPOMI analysis. EM and AR provided data from OMPS-LP. CvS provided CALIOP data. FJP retrieved and analyzed AHI data. LOM and GAH prepared the manuscript with significant contributions from all authors.

*Competing interests.* The authors declare that they have no conflict of interest.

*Acknowledgements.* The main contribution to this work by LOM has been funded under FE-Nr. 50.0368/2017 by BMVI (Federal Ministry of Transport and Digital Infrastructure of Germany). The contributions from GAH, AH, and JB are within the VolPlume project as a part of the research unit VolImpact funded by German research foundation DFG (FOR 2820). As part of the same research unit (VolImpact), EM, AR, and CvS contributed within the VolARC project and SW within VolDyn, respectively.

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
