# Peer review of "Particle Aging and Aerosol–Radiation Interaction Affect Volcanic Plume Dispersion: Evidence from Raikoke Eruption 2019"

_Atmospheric Chemistry and Physics, 2020_

## Referee Comment (RC1) · Arnau Folch (Referee) · 14 Jul 2020

This paper uses the ICON-ART modelling system to study the effects of volcanic aerosol dynamics (alterations in aerosol size and composition due to particle aging) and aerosol-radiation feedbacks on the dynamics of volcanic clouds. It is known that the strong absorption of fine ash particles can cause thermal disequilibrium with the surrounding atmosphere, potentially altering the atmospheric dynamics. However, in-depth studies are scarce in the literature and this paper is an important step forward. The authors show results for the 2019 Raikoke eruption, using measurements from different satellite instrumentation for model validation; TROPOMI and AHI for $SO_2$/ash

column mass retrievals, and MOIS/VIIRS/CALIOP/OMPS-LP for cloud top height. It is difficult to extract conclusions from a single example but, overall, I think this paper is very relevant to show the potential effects of both phenomena on model forecasts. I do recommend publication with minor revisions detailed below.

- ICON-ART is run for 3 scenarios: AERODYN_rad (aerosol dynamics + radiation), no_AERODYN_rad (no dynamics) and AERODYN_no_rad (no radiation), which allow isolating the effects of dynamics and radiation. These are actually in competition, with dynamics enhancing premature settling and radiation uplifting the cloud (as nicely shown in Figure 8). To what extent can these two effects counterbalance? This is somehow discussed in Sec 3.3., but it would be great to compare AERODYN-rad results with the no_AERODYN_no_rad ICON case. Note also that, to my knowledge, all operational volcanic cloud forecast systems do not include neither dynamics nor radiation and therefore the no_AERODYN_no_rad (not shown) would actually mimic current setups.

- Figure 4 is very interesting but panels (c)-(e) (and (d)-(f)) are difficult to distinguish and should highlight differences better (e.g. using a log scale). A better option could be plotting relative differences (in percent) between both model configurations, using AERODYN_rad as the "true". Is it a 10\% or a 100\%? Difficult to say from (d)-(f) with the contour range used.

- On the other hand, and related to the point above, I missed some figure or text showing the impact on the atmospheric dynamics when switching on the AERODYN_rad module. To what extent is the vertical wind field advecting the cloud modified by thermal perturbations? Can you quantify? I understand that this question may fall beyond the objective of the paper, but it could be of interest to the volcanic cloud modelling community. Ensemble forecast strategies are gaining more and more attention, and these rely on perturbing uncertain variables like the eruption source parameters or the wind field (but rarely the vertical component). As a result, an interesting question it to assess whether (vertical) wind perturbations caused by radiation feedbacks are comparable to typical uncertainties in NWP models. If in the range, an ensemble of off-line models could still capture this effect, at least to some extent.

- The aerosol dynamics module (ARODYN) has pre-defined initial aerosol size distributions, which (if I am not wrong) are evolved according to prognostic equations. How does the aging mechanism depend on this initial condition? Particle distributions can vary notably from one eruption to another, and a single representation could be misleading.

- Model validation. Several plots compare model results with observations. However, I missed some quantitative metric values; e.g. SAL, Figure Merit of Space or others. These are by far more objective than color plots (e.g. Figs 4, 5), which can trick depending on the scale and color binning. Given that a main objective of the paper is to "assess if representations of aerosol dynamics and aerosol-radiation interactions are beneficial for forecasts", quantitative metrics would help asking this question more objectively.

- Line 84. "density values less"?

- Line 257. It is stated that the source term in ICON-ART is set between 8 and 14 km a.s.l. Does it mean a 6 km thick cloud? This seems quite inconsistent with the TROPOMI retrievals, which assume 1 km thickness at 15 km a.s.l.

---

## Referee Comment (RC2) · Anonymous Referee #1 · 16 Jul 2020

In this study, the authors investigate the importance of aerosol dynamics and aerosol-radiation interactions in the early dispersion of the volcanic plume injected by the Raikoke eruption in June 2019. They argue that physical processes influencing the transport of volcanic plumes in the UTLS region have been poorly addressed compared to work related to source parameters/initial conditions. Using a set of satellite observations including HIMAWARI-8, CALIOP and OMPS-LP, they attempt to validate their simulations of the ICON-CART global modelling system. This is a very interesting and unique study that attempt to shed light on how a complex aerosol-dynamic-radiation coupling system can be used to understand early evolution of volcanic plumes and thus is suitable for publication in the Atmospheric Chemistry and Physics Journal. However,

[Figure]

I believe that additional work would need to be done to validate the model results. With only one CALIPSO browse image and one OMPS-LP volcanic plume top point, the vertically resolved information that offer a unique opportunity to validate model results are not fully explored. Before this manuscript can be published, I would recommend the authors to provide additional observational evidences to support their conclusions. Herein below are additional comments that the authors may want to consider:

P1L3: I agree with this statement but essential information about mass injection rates and plume injection heights are still critical parameters to simulate volcanic plume dispersion.

P1L10: I would replace "show" by "suggest" since I'm not certain that the results presented in this paper really fully support the conclusions.

P2L36: I would argue that the rise of the plume is better documented by the two initial papers from Khaykin et al., 2017 and Peterson et al., 2017.

P3L83: Could you explain what's the implications of selecting qa_value larger than 0.5 ?

P4L109: One sentence about the adjustment technique could be explained here.

P5L126: What could be the impact of ice on those estimates?

P6L167: This is very unlikely that the Ambae eruption had a significant impact on stratospheric aerosols beyond the tropics and sub-tropics and thus it seems unrealistic to consider that Ambae could impact the retrieval of a fresh volcanic plume within the OMPS data set within the latitude band where the Raikoke was transported during the first few days.

P9L240: The treatment of externally mixed ash and sulfuric acid would be more accurate through T-Matrix calculation than Mie Theory. I think this could be further discuss in the manuscript since it seems to be an important element.

P15L349: The other optical properties (depolarization/color ratio/vertical feature mask) from the plumes from CALIPSO are not shown. This would certainly help with the interpretation as well.

Figure 6: Does the model really do a better job representing the volcanic plume with the full dynamical-chemistry-radiation coupling? I'm not really certain that the figure demonstrate that since pieces of plume seen by the AERODYN-rad scenario do not appear clearly on the observations. See link to CALIPSO browse image crossing the volcanic cloud on Jun 22nd for additionnal obs. that could be used to validate model results. https://www-calipso.larc.nasa.gov/products/lidar/browse_images/show_v4_detail.php?s=production&v=V4-10&browse_date=2019-06-22&orbit_time=01-59-01&page=3&granule_name=CAL_LID_L1-Standard-V4-10.2019-06-22T01-59-01ZD.hdf.

Figure 7: Even if the model indeed do a better job by including the dynamics and radiation to remove ash, it does not capture well small-scale variations. Could you further explain why it's not the case? Maybe incorporating more accurate source terms based on HIMAWARI-8 would help with that.

P17L375: It would be interesting to know which processes contribute to the removal of ash in the model. I believe the growth term that lead to the removal by sedimentation, what about ash-ice interaction and wet deposition ?

Figure 9: More data are needed to verify the model outputs. e.g. CALIPSO and OMPS.

P20L431: I believe that measurement uncertainty from OMPS could be better addressed. The vertical resolution of the instrument is probably near 1-2 km. Could you add the corresponding error bar in figure 8. In addition, I'm pretty confident that additional information on volcanic cloud top height could be found by analyzing additional OMPS data.

---

## Author Comment (AC1) · 22 Sep 2020

Dear Dr Folch,

We thank you a lot for your valuable comments and suggestions. We addressed them as explained below.

The reviewer's comments are repeated in **bold letters**, our replies are given in standard font, and text modified or added to the manuscript is given in blue.

**This paper uses the ICON-ART modelling system to study the effects of volcanic aerosol dynamics (alterations in aerosol size and composition due to particle aging) and aerosol-radiation feedbacks on the dynamics of volcanic clouds. It is known that the strong absorption of fine ash particles can cause thermal disequilibrium with the surrounding atmosphere, potentially altering the atmospheric dynamics. However, in-depth studies are scarce in the literature and this paper is an important step forward. The authors show results for the 2019 Raikoke eruption, using measurements from different satellite instrumentation for model validation; TROPOMI and AHI for SO2/ash column mass retrievals, and MOIS/VIIRS/CALIOP/OMPS-LP for cloud top height. It is difficult to extract conclusions from a single example but, overall, I think this paper is very relevant to show the potential effects of both phenomena on model forecasts. I do recommend publication with minor revisions detailed below.**

Thank you very much for the insightful review. Your comments and questions helped us a lot to improve the manuscript.

**1. ICON-ART is run for 3 scenarios: AERODYN_rad (aerosol dynamics + radiation), no_AERODYN_rad (no dynamics) and AERODYN_no_rad (no radiation), which allow isolating the effects of dynamics and radiation. These are actually in competition, with dynamics enhancing premature settling and radiation uplifting the cloud (as nicely shown in Figure 8). To what extent can these two effects counterbalance? This is somehow discussed in Sec 3.3., but it would be great to compare AERODYN-rad results with the no_AERODYN_no_rad ICON case. Note also that, to my knowledge, all operational volcanic cloud forecast systems do not include neither dynamics nor radiation and therefore the no_AERODYN_no_rad (not shown) would actually mimic current setups.**

We agree that operational volcanic cloud forecast centers do neither include dynamics nor radiation interaction. Therefore, a comparison with this simulation case would indeed be beneficial. As we ran ICON-ART in the setup no_AERODYN-no_rad, we add some of these results to the manuscript.

Updated Table 2:
We include the no_AERODYN-no_rad scenario in Table 2.

We add to l. 279:
The fourth scenario represents the status quo of operational volcanic cloud forecast. It considers neither aerosol dynamic effects nor aerosol-radiation interaction.

Updated Fig. 4:
We replace the AERODYN-no_rad by the no_AERODYN-no_rad scenario. For details, please refer to answer of comment 2.

Updated Fig. 6:
We include the two no_rad simulation scenarios in Fig. 6. Furthermore, additional dates with a comparison between CALIOP and ICON-ART model results are displayed in the Appendix of the manuscript.
We add to l. 358:
A similar conclusion can be derived from the AERODYN-no_rad and no_AERODYN-no_rad scenarios in Fig. 6 (e) and (f), respectively. Although, both are missing the most prominent feature between 49° N and 51° N at around 16 km, they show the same behavior in terms of aerosol dynamic effects.
Additional dates of CALIPSO measurements are displayed in Appendix A.

[Figure]

Figure 6. (a) CALIPSO ground track on 23 June 2019, around 15:00 UTC in blue color and location of Raikoke volcano as red triangle. The contour map shows the volcanic ash cloud top height for the AERODYN-rad scenario. (b) The CALIOP attenuated backscatter for 532 nm for the satellite position between 40° N and 70° N is displayed in the top right panel. The magenta line shows the 0.002 km⁻¹sr⁻¹ contour of AERODYN-rad at 15:00 UTC. Middle and lower panels: Total attenuated backscatter for 532 nm of volcanic aerosols under the CALIPSO ground track on 23 June 2019, for the 15:00 UTC model output are displayed. (c) shows the result for AERODYN-rad, (d) for no_AERODYN-rad, (e) for AERODYN-no_rad, and (f) for no_AERODYN-no_rad, respectively.

Updated Fig. 8:
We include the no_AERODYN-no_rad scenario plume top height in Fig. 8. Furthermore, we add an error bar for the OMPS measurement in the same figure (as requested by referee #1).
For further explanation we rephrase l. 410:
A distinct difference prevails between the two scenarios with radiative interaction (yellow and green curve) and the two without radiative interaction (pink and orange curve).

And add to l. 420:
As for accumulation mode particles, in the no_AERODYN-no_rad scenario (orange curve) coarse mode particles also tend to stay on the same height level.

[Figure]

Figure 8. (a) and (b) Evolution of height of volcanic ash cloud top after the onset of the eruption on 21 June 2019, at 18:00 UTC. The yellow curve represents the no_AERODYN-rad scenario, the green curve AERODYN-rad, the pink one AERODYN-no_rad, and the orange one represents the no_AERODYN-no_rad scenario. Panel (a) shows the ash cloud top of particles in the accumulation mode, (b) of particles in the coarse mode, respectively. The black circle depicts the volcanic cloud top height obtained from OMPS-LP. (c) Mean temperature difference (AERODYN-rad – AERODYN-no_rad) in volcanic ash cloud columns on 23 June 2019, 12:00 UTC. (d) Mean volcanic ash concentration $\bar{\chi}$ for the same model columns as in (c) for AERODYN-rad.

**2. Figure 4 is very interesting but panels (c)-(e) (and (d)-(f)) are difficult to distinguish and should highlight differences better (e.g. using a log scale). A better option could be plotting relative differences (in percent) between both model configurations, using AERODYN_rad as the "true". Is it a 10\% or a 100\%? Difficult to say from (d)-(f) with the contour range used.**

We updated Fig. 4 in two ways. First of all, we replaced panels (e) and (f) by the total column ash mass loading of the no_AERODYN-no_rad case. Secondly, we added panels (g) and (h) which are showing the absolute difference between the two simulation scenarios AERODYN-rad – no_AERODYN-no_rad.

We add to and rephrase l. 293:
These differences are mainly restricted to the slightly higher mass loading in panel (e) and small differences in the volcanic cloud structure. For the first day after the eruption, the aerosol dynamic effects and the aerosol-radiation interaction have only a minor influence on the volcanic ash mass loading.
We add to and rephrase l. 300ff.:
Compared to these two simulations, the averaged AHI measurements (Fig. 4 (b)) show values for the maximum ash mass loading that lie in between the two simulation scenarios. In panels (g) and (h) the differences between the two are highlighted by the absolute difference of AERODYN-rad – no_AERODYN-no_rad. It shows that considering aerosol dynamics and aerosol-radiation interaction results in lower volcanic ash mass loadings in most parts of the volcanic cloud. Only for the first day after the eruption, the volcanic cloud seems to be shifted slightly north in the AERODYN-rad scenario compared to the no_AERODYN-no_rad scenario, as the difference plot shows some positive values between 160 – 170° N.

[Figure]

Figure 4. Daily mean total column mass loading of volcanic ash on 22 June (left column) and 23 June 2019 (right column). Top row (panel (a) and (b)) shows results measured by AHI on-board Himawari-8. The middle and lower row (panel (c) - (f)) show ICON-ART results for AERODYN-rad and no_AERODYN-no_rad, respectively. The black triangle depicts the location of Raikoke volcano. Panels (g) and (h) show the absolute difference between the two simulation scenarios.

**3. On the other hand, and related to the point above, I missed some figure or text showing the impact on the atmospheric dynamics when switching on the AERODYN_rad module. To what extent is the vertical wind field advecting the cloud modified by thermal perturbations? Can you quantify? I understand that this question may fall beyond the objective of the paper, but it could be of interest to the volcanic cloud modelling community. Ensemble forecast strategies are gaining more and more attention, and these rely on perturbing uncertain variables like the eruption source parameters or the wind field (but rarely the vertical component). As a result, an interesting question it to assess whether (vertical) wind perturbations caused by radiation feedbacks are comparable to typical uncertainties in NWP models. If in the range, an ensemble of offline models could still capture this effect, at least to some extent.**

If we look at the most pronounced lifting of the volcanic cloud top height (approx. 3 km, compare Fig. 8) during the first 12 h of the simulation, we obtain an average vertical lifting velocity of 0.07 m/s. This lifting is only visible for simulation scenarios with radiation interaction.
We determined the vertical velocity difference between the AERODYN-rad and the no_AERODYN-no_rad scenario as well as between the AERODYN-rad and AERODYN-no_rad scenario. Both comparisons show comparable numbers. For the comparison, we only consider grid cells which are within a vertical column which contains a volcanic ash mass loading > 0.01 g m$^{-2}$. The maximum absolute difference that appears locally during the first 12 h of the eruption is in the order of 0.19 m/s with a 98$^{th}$ percentile of 0.05 m/s. We would like to note, that these vertical velocity perturbations strongly depend on the spatial resolution. For a finer resolution, locally we would expect higher vertical velocities.

We include this information in the manuscript in l. 431ff.:
The resulting vertical velocity perturbation $\Delta w$ is in the order of 0.1 m s$^{-1}$. For this purpose, we analyzed the difference in vertical velocity between the AERODYN-rad and AERODYN-no_rad scenario during the first 12 h after the eruption. Only grid cells in model columns which contain a volcanic ash mass loading > 0.01 g m$^{-2}$ in both scenarios are considered. Locally, $\Delta w$ reaches 0.19 m s$^{-1}$ with a 98th percentile of 0.05 m s$^{-1}$. This agrees well with the vertical lifting of the volcanic cloud top height of around 3 km during the first 12 h ($\overline{w}$ = 0.07 m s$^{-1}$).

**4. The aerosol dynamics module (ARODYN) has pre-defined initial aerosol size distributions, which (if I am not wrong) are evolved according to prognostic equations. How does the aging mechanism depend on this initial condition? Particle distributions can vary notably from one eruption to another, and a single representation could be misleading.**

We agree that particle size distributions (PSDs) can vary notably from one eruption to another. For the Raikoke simulation, we defined the emitted PSD as specified in Table 1 in the manuscript. This emitted PSD changes over time as particles age and sediment. Very often, there is lack of direct measurements when it comes to the PSD of volcanic ash from one particular volcanic eruption like Raikoke. To overcome this limitation, we used the PSD data from five eruptions (listed in the following table) to calculate a generic PSD for volcanic ash in ICON-ART (shown in the following figure). This PSD captures the variability of fine and coarse particles. We are aware of the uncertainties associated with

this generic PSD. Nevertheless, even direct measurements are subjected to large uncertainties and might fail to represent the variability of the PSD.

The aging mechanism which is implemented in AERODYN depends on the PSD. As condensation of gaseous species on existing particles, coagulation, sedimentation, and deposition directly depend on the particle diameter. However, it needs further studies in order to quantify how the aging mechanism depends on the emitted size distribution.

Table: Volcanic eruption for which validated ash PSD data exist (from http://www.ct.ingv.it/iavcei/results.htm)

| Eruption | Montserrat (West Indies)

31 March 1997 | Mt. St. Helens (USA)

18 May 1980 | Ruapehu (New Zealand)

17 June 1996 | Spurr (Alaska)

16-17 September 1992 | Eyjafjallajökull (Iceland)

14 April-21 May 2010

(data for 4-8 May 2010) |
|---|---|---|---|---|---|
| Eruption type | dome collapse

(co-PF plumes) | Plinian+coignimbrite

(strong plume) | sub-plinian

(weak plume) | sub-plinian | long-lasting weak plume |

[Figure]

Figure: Calculated ash PSD based on the data available in literature

Bonadonna, C. and Scollo, S.: IAVCEI Commission on Tephra Hazard Modelling, http://www.ct.ingv.it/iavcei/results.htm, last access: 03 September 2020, 2013.

We add to l. 273:
They are based on data from Bonadonna and Scollo (2013).

**5. Model validation. Several plots compare model results with observations. However, I missed some quantitative metric values; e.g. SAL, Figure Merit of Space or others. These are by far more objective than color plots (e.g. Figs 4, 5), which can trick depending on the scale and color binning. Given that a main objective of the paper is to "assess if representations of aerosol dynamics and aerosol-radiation interactions are beneficial for forecasts", quantitative metrics would help asking this question more objectively.**

We apply the SAL method following Wernli et al. (2008) in order to compare the total column volcanic ash mass loading AHI retrieval with our model result.

Wernli, H., M. Paulat, M. Hagen, and C. Frei, 2008: SAL—A Novel Quality Measure for the Verification of Quantitative Precipitation Forecasts. Mon. Wea. Rev., 136, 4470–4487, https://doi.org/10.1175/2008MWR2415.1

Wernli, H., C. Hofmann, and M. Zimmer, 2009: Spatial Forecast Verification Methods Intercomparison Project: Application of the SAL Technique. Wea. Forecasting, 24, 1472–1484, https://doi.org/10.1175/2009WAF2222271.1

We add the following paragraph to the manuscript in l. 302ff.:
In order to compare our ICON-ART results in an objective manner with the AHI observations, we make use of the SAL method. This quality measure has been introduced by Wernli et al. (2008) and has been extensively discussed by Wernli et al. (2009). The method identifies objects in a 2D field (e.g., total ash mass loading) and quantifies the differences between model and observation in structure (S), amplitude (A), and location (L). A value of 0 implies perfect agreement. We apply the SAL method with a fix threshold value to identify objects R* = 0.01 g m$^{-2}$. The results for the comparison of daily mean total column mass loading between the AHI retrieval and the ICON-ART results are summarized in Table 3.
The location of the volcanic cloud agrees very well with the observation for all dates in all simulation scenarios. The structure of the volcanic cloud shows larger differences compared to observations, especially on 23 June. However, the values are rather similar for the different simulation scenarios. Only the amplitude values differ distinctly among the different scenarios. Simulations with AERODYN are closer to the observation than simulations without aerosol dynamics.

Table 3: Comparison of daily mean total column mass loading of volcanic ash between AHI and ICON-ART results using the SAL method by Wernli et al. (2008).

| scenario | 2019-06-22 | | | 2019-06-23 | | |
|---|---|---|---|---|---|---|
| | S | A | L | S | A | L |
| AERODYN-rad | −0.191 | 0.584 | 0.004 | 1.651 | 0.298 | 0.041 |
| AERODYN-no_rad | −0.323 | 0.579 | 0.002 | 1.362 | 0.275 | 0.028 |
| no_AERODYN-rad | −0.202 | 0.921 | 0.014 | 1.601 | 0.716 | 0.031 |
| no_AERODYN-no_rad | −0.270 | 0.874 | 0.013 | 1.546 | 0.748 | 0.030 |

**6. Line 84. "density values less"?**

We agree that this formulation is a bit misleading and hope that the reformulation makes it easier to understand.
We change the sentence on p.3 l.84 from:
Only data with the quality descriptor 'qa_value' larger than 0.5 and total vertical column density values less than 1000 mol m$^{-2}$ were used.

to:

Only data with a quality value larger than 0.5 (as recommended in the TROPOMI product user manual) and total vertical column density with values less than 1000 mol m$^{-2}$ were used.

**7. Line 257. It is stated that the source term in ICON-ART is set between 8 and 14 km a.s.l. Does it mean a 6 km thick cloud? This seems quite inconsistent with the TROPOMI retrievals, which assume 1 km thickness at 15 km a.s.l.**

Yes, in the model simulation we emit a 6km thick cloud of ash and $SO_2$. Our emission parametrization for ash and $SO_2$ is based on satellite observations (as well as results of Plumeria and FPlume). The configuration of the emission height has been done specifically for the Raikoke eruption in 2019 and is based on satellite observations and volcanic monitoring reports (Sennet, 2019).

Whereas, the TROPOMI retrieval assumptions have been set for a much broader range of scenarios. The retrieval algorithm can be run with one of four different assumptions on where the $SO_2$ is located in the atmosphere. This could either be a vertical profile modeled by the global chemistry transport model TM5 or a 1 km thick box in either 1 km, 7 km or 15 km. Comparisons with other satellite products showed, that the assumption of a 1 km box in 15 km a.s.l. gave best results, although, the retrieval assumption does not match with the actual $SO_2$ distribution in the atmosphere after the Raikoke eruption.

---

## Author Comment (AC2) · 22 Sep 2020

Dear Referee 1,
We thank you a lot for your valuable comments and suggestions. We addressed them as explained below.
The reviewer's comments are repeated in **bold letters**, our replies are given in standard font, and text modified or added to the manuscript is given in blue.

**In this study, the authors investigate the importance of aerosol dynamics and aerosol-radiation interactions in the early dispersion of the volcanic plume injected by the Raikoke eruption in June 2019. They argue that physical processes influencing the transport of volcanic plumes in the UTLS region have been poorly addressed compared to work related to source parameters/initial conditions. Using a set of satellite observations including HIMAWARI-8, CALIOP and OMPS-LP, they attempt to validate their simulations of the ICON-CART global modelling system. This is a very interesting and unique study that attempt to shed light on how a complex aerosol-dynamic-radiation coupling system can be used to understand early evolution of volcanic plumes and thus is suitable for publication in the Atmospheric Chemistry and Physics Journal. However, I believe that additional work would need to be done to validate the model results. With only one CALIPSO browse image and one OMPS-LP volcanic plume top point, the vertically resolved information that offer a unique opportunity to validate model results are not fully explored. Before this manuscript can be published, I would recommend the authors to provide additional observational evidences to support their conclusions.**

Thank you very much for the insightful review. Your comments and questions helped us a lot to improve the manuscript.
We agree that additional observational data, especially in form of OMPS-LP volcanic cloud top height, would be very beneficial for the validation of the model results. Unfortunately, we were not able to retrieve any meaningful volcanic cloud height from OMPS-LP measurements for other dates. The reason for this is discussed in more detail together with the answer to comment 7.
The CALIOP measurements show a signal that can be associated with the volcanic cloud on other dates as well. We included these in our answers to the respective comments.

**1. P1L3: I agree with this statement but essential information about mass injection rates and plume injection heights are still critical parameters to simulate volcanic plume dispersion.**

Yes, we totally agree with you that the correct representation of source parameters is very critical for a reliable forecast of volcanic aerosols. Especially, estimates about the mass eruption rate and plume height are crucial for short term forecasts right after volcanic eruptions. This is why they are still substance of ongoing research. With this work, we don't intend to diminish the importance of source parameters, but shed light on less studied sink processes.

**2. P1L10: I would replace "show" by "suggest" since I'm not certain that the results presented in this paper really fully support the conclusions.**

During the review process we had the opportunity to provide further evidence for our statement. This is why we would like to leave it as is.

**3. P2L36: I would argue that the rise of the plume is better documented by the two initial papers from Khaykin et al., 2017 and Peterson et al., 2017.**

We additionally cite the suggested two papers:

Khaykin, S. M., Godin-Beekmann, S., Keckhut, P., Hauchecorne, A., Jumelet, J., Vernier, J.-P., Bourassa, A., Degenstein, D. A., Rieger, L. A., Bingen, C., Vanhellemont, F., Robert, C., DeLand, M., and Bhartia, P. K.: Variability and evolution of the midlatitude stratospheric aerosol budget from 22 years of ground-based lidar and satellite observations, Atmos. Chem. Phys., 17, 1829–1845, https://doi.org/10.5194/acp-17-1829-2017, 2017.

Peterson, P. K., Pöhler, D., Sihler, H., Zielcke, J., General, S., Frieß, U., Platt, U., Simpson, W. R., Nghiem, S. V., Shepson, P. B., Stirm, B. H., Dhaniyala, S., Wagner, T., Caulton, D. R., Fuentes, J. D., and Pratt, K. A.: Observations of bromine monoxide transport in the Arctic sustained on aerosol particles, Atmos. Chem. Phys., 17, 7567–7579, https://doi.org/10.5194/acp-17-7567-2017, 2017.

We add to the manuscript in l. 31:
This can result in a lofting mechanism of aerosol which is different from the one caused by large scale atmospheric dynamics as described for example by Khaykin et al. (2017).

In l. 34:
Peterson et al. (2017) observed in the Arctic near-surface atmosphere that the transport of atmospheric pollutants is influenced by active halogen chemistry.

**4. P3L83: Could you explain what's the implications of selecting qa_value larger than 0.5?**

The qa_value is described in the ESA Tropomi User Manual as followed:
"The quality value or qa_value is a continuous quality descriptor, varying between 0 (no data) and 1 (full quality data). Recommend to ignore data with qa_value < 0.5 (static)" (Sentinel-5 precursor/TROPOMI Level 2 Product User Manual Sulphur Dioxide SO2, https://sentinel.esa.int/documents/247904/2474726/Sentinel-5P-Level-2-Product-User-Manual-Sulphur-Dioxide , accessed 23 July 2020)
In order to improve comprehensibility, we reformulate P3L83
Only data with the quality descriptor 'qa_value' larger than 0.5 and total vertical column density values less than 1000 mol m$^{-2}$ were used.

to:
Only data with a quality value larger than 0.5 (as recommended in the TROPOMI product user manual) and total vertical column density with values less than 1000 mol m$^{-2}$ were used.

**5. P4L109: One sentence about the adjustment technique could be explained here.**

We rephrase and add some extra information in l.109.
Water vapor and clouds cause interference with the $SO_2$ signal and introduce a positive bias. Therefore, a retrieval scheme was devised to minimize the interfering effects. In short, the bias is minimized by subtracting an offset $SO_2$ retrieval for a small region where no $SO_2$ is believed to exist.

**6. P5L126: What could be the impact of ice on those estimates?**

We thank the reviewer for raising this as it is a very good point and one that we should have addressed in the manuscript. Ice formation in volcanic clouds is a known problem and happens often, especially in water-rich and tropical eruptions where moist air entrainment happens see Prata et al. (2020). Ice has a very clear infrared spectral signature that can be used to diagnose its presence in volcanic clouds. For Raikoke this signature was absent or at best, weak. True-color images from the Himawari-8 satellite also show no obvious signs of ice - the clouds are dark brown and become paler with time, presumably because of dispersion. The absence of an ice signature can be explained by the high altitude of the emissions (>8 km and up to 15 km) which deposited them into a very dry part of the atmosphere, and the lack of a water-rich plume to begin with, as evidenced in the true-color and IR spectral signature data. The presence of ice reduces the ash mass estimates by an amount that depends on the proportion of the pixel covered by ice. Ice formation could have occurred in the early (first few hours of the eruption on 21 June) as the plume ascended through a moister part of the atmosphere. This could partly explain why ash estimates at the start of the eruption are low; but ash opacity is also a factor that reduces the ash mass retrieval.

Prata, A.T., Folch, A., Prata, A.J., Biondi, R., Brenot, H., Cimarelli, C., Corradini, S., Lapierre, J. and Costa, A., 2020. Anak Krakatau triggers volcanic freezer in the upper troposphere. Scientific reports, 10(1), pp.1-13.

In order to address this issue, we add the following to the manuscript at l.120:

The presence of ice reduces the ash mass estimates by an amount that depends on the proportion of the pixel covered by ice. However, during the Raikoke eruption, ice was not observed except possibly at the start of the eruption which could cause lower ash mass estimates.

**7. P6L167: This is very unlikely that the Ambae eruption had a significant impact on stratospheric aerosols beyond the tropics and sub-tropics and thus it seems unrealistic to consider that Ambae could impact the retrieval of a fresh volcanic plume within the OMPS data set within the latitude band where the Raikoke was transported during the first few days.**

As the study of Malinina et al. (2020, in review at ACP) on the Ambae eruption shows, the Ambae plume spreads up to 40N until the end of 2018, where its influence still remains non-negligible. Thus, influence of the Ambae eruption at 50N in June 2019 might be expected. We agree with the reviewer that these small residual signals do not affect the retrievals in the core of the fresh Raikoke plume. Unfortunately, the sampling of the OMPS-LP instrument is quite sparse and it does not hit the core of the fresh plume on other days than the one analyzed in the paper. During the analyzed period, in the transition regions of the plume the increase of the aerosol extinction measured by OMPS-LP was not that pronounced and thus can be interfered by residual signals from previous events. We change the manuscript to make this clearer.

Malinina, E., Rozanov, A., Niemeier, U., Peglow, S., Arosio, C., Wrana, F., Timmreck, C., von Savigny, C., and Burrows, J. P.: Changes in stratospheric aerosol extinction coefficient after the 2018 Ambae eruption as seen by OMPS-LP and ECHAM5-HAM, Atmos. Chem. Phys. Discuss., https://doi.org/10.5194/acp-2020-749, in review, 2020

We additionally cite Malinina et al. (2020) in l. 161:
Detailed information on the retrieval algorithm can be found in Malinina (2019) and Malinina et al. (2020).

We rephrase l. 165ff.:
In the following days, when the plume started to spread over the North Pacific, the core of the fresh plume is not hit by the OMPS-LP instrument sampling anymore. Slightly perturbed aerosol extinction observed in transition regions has a similar magnitude as that from interfering events, e.g., the aerosol transport from the Ambae eruption that occurred 11 months earlier, and thus cannot be attributed exclusively to the Raikoke eruption. For this reason, we excluded the OMPS-LP measurements in transition regions from the consideration.

**8. P9L240: The treatment of externally mixed ash and sulfuric acid would be more accurate through T-Matrix calculation than Mie Theory. I think this could be further discuss in the manuscript since it seems to be an important element.**

We agree with the reviewer that T-Matrix calculations are a powerful tool to determine the radiation interaction of non-spherical aerosols, such as volcanic ash. However, in this work in combination with the newly developed AERODYN module, we allow the formation of internally mixed particles, such as volcanic ash coated with a shell of sulfate and water. To our knowledge there is no T-Matrix code that can handle core-shell assumption. That is why we make use of Mie Theory assuming a core-shell mixing state.
Due to the coating, a spherical assumption for these mixed particles might be reasonable. It is only for consistency reasons, why we chose to apply the sphericity assumption also to the uncoated ash particles. Implementing coated non-spherical ash particles into ICON-ART or considering the non-sphericity of uncoated particles together with internally mixed ones remains the subject of future work.

**9. P15L349: The other optical properties (depolarization/color ratio/vertical feature mask) from the plumes from CALIPSO are not shown. This would certainly help with the interpretation as well.**

We agree that other optical properties retrieved from CALIOP measurements, such as depolarization ratio, give beneficial information about the composition of the aerosol cloud. In the current state of ICON-ART we don't have forward operators for these quantities. However, to our knowledge it is the first model that retrieves the total attenuated backscatter for internally mixed volcanic aerosol. This is why in this paper, we only compare the total attenuated backscatter at 532nm.

In the manuscript we argue that model results could help to interpret observations better. As an example, we took two images of the CALIOP Aerosol Subtype classification of two dates when the satellite passed over the volcanic cloud. In both images the blue rectangle highlights an area where the plume is located in our model result and also shows a signal in the total attenuated backscatter measured by CALIOP. For the first date, the detected aerosol (within the blue box) is classified only partly as volcanic ash. Based on our model result for the same date we would argue, that in fact the here classified dust is volcanic ash as well. For the other image, there is no aerosol type classified within the blue box. However, the total attenuated backscatter clearly shows a signal and our model results suggest that this is indeed volcanic ash.

[Figure]

[Figure]

**10. Figure 6: Does the model really do a better job representing the volcanic plume with the full dynamical-chemistry-radiation coupling? I'm not really certain that the figure demonstrate that since pieces of plume seen by the AERODYN-rad scenario do not appear clearly on the observations. See link to CALIPSO browse image crossing the volcanic cloud on Jun 22nd for additional obs. that could be used to validate model results. https://www-calipso.larc.nasa.gov/products/lidar/browse_images/show_v4_detail.php?s=production&v=V4-10&browse_date=2019-06-22&orbit_time=01-59-01&page=3&granule_name=CAL_LID_L1-Standard-V4-10.2019-06-22T01-59-01ZD.hdf**

We evaluated four additional dates for which CALIPSO passes over the volcanic cloud of the Raikoke 2019 eruption. Furthermore, we extended the model comparison by the two no-rad simulation scenarios as requested by reviewer Arnau Folch. These plots will be displayed in the appendix of the manuscript.

The evaluation of these additional dates confirms our previous statement regarding the improvement of the forecast by including aerosol dynamics and radiation interactions. Only the very last date on June 25 shows no significant improvement.

[Figure]

*2019-06-22 03:00 UTC*
*(a) CALIPSO ground track and modeled volcanic cloud top height*
*(b) Total Attenuated Backscatter at 532 nm measured by CALIOP*
*(c) AERODYN – rad*
*(d) no AERODYN – rad*
*(e) AERODYN – no rad*
*(f) no AERODYN – no rad*

[Figure]

2019-06-23 02:00 UTC
(a) CALIPSO ground track and modeled volcanic cloud top height
(b) Total Attenuated Backscatter at 532 nm measured by CALIOP
(c) AERODYN – rad
(d) no AERODYN – rad
(e) AERODYN – no rad
(f) no AERODYN – no rad

[Figure]

Figure 6. (a) CALIPSO ground track on 23 June 2019, around 15:00 UTC in blue color and location of Raikoke volcano as red triangle. The contour map shows the volcanic ash cloud top height for the AERODYN-rad scenario. (b) The CALIOP attenuated backscatter for 532 nm for the satellite position between 40° N and 70° N is displayed in the top right panel. The magenta line shows the 0.002 km$^{-1}$sr$^{-1}$ contour of AERODYN-rad at 15:00 UTC. Middle and lower panels: Total attenuated backscatter for 532 nm of volcanic aerosols under the CALIPSO ground track on 23 June 2019, for the 15:00 UTC model output are displayed. (c) shows the result for AERODYN-rad, (d) for no_AERODYN-rad, (e) for AERODYN-no_rad, and (f) for no_AERODYN-no_rad, respectively.

[Figure]

*2019-06-24 16:00 UTC*
*(a) CALIPSO ground track and modeled volcanic cloud top height*
*(b) Total Attenuated Backscatter at 532 nm measured by CALIOP*
*(c) AERODYN – rad*
*(d) no AERODYN – rad*
*(e) AERODYN – no rad*
*(f) no AERODYN – no rad*

[Figure]

*2019-06-25 01:00 UTC*
*(a) CALIPSO ground track and modeled volcanic cloud top height*
*(b) Total Attenuated Backscatter at 532 nm measured by CALIOP*
*(c) AERODYN – rad*
*(d) no AERODYN – rad*
*(e) AERODYN – no rad*
*(f) no AERODYN – no rad*

**11. Figure 7: Even if the model indeed do a better job by including the dynamics and radiation to remove ash, it does not capture well small-scale variations. Could you further explain why it's not the case? Maybe incorporating more accurate source terms based on HIMAWARI-8 would help with that.**

This is a very good point. Although the overall agreement is very good, there are small-scale variations in the AHI retrieval that don't have a corresponding model result.
At this point we should distinguish between two periods, the eruption period, roughly during the first 12h, and the quiet period during which the volcano did not emit ash anymore.
In this study we emit volcanic ash with a constant emission rate over 9h. We know from satellite images (GEOS17, Himawari-8) that Raikoke emitted ash with several longer and shorter puffs. The last, rather short puff happened on 22 July 2019 at around 07:10 UTC. This explains well the offset between model result and observation, and also the small-scale variations in the observation during the eruption phase. Characterizing these puffs in terms of height and mass eruption rate (and thus time dependent eruption rate) is the topic of ongoing work.
After the eruption has stopped, the small-scale variations in the AHI retrieval are due to deficiencies or limitations of the retrieval algorithm, as any increase of measured ash cannot be associated with an emission.
In order to make this clearer in the manuscript, we add additional information to l.368.

As Raikoke did not erupt continuously over these 9 h, the offset between simulation and observation as well as the small-scale variations in the observation during this period can be explained.

Furthermore, we add one and rephrase one sentence in l. 373:
The small-scale variations in the observation might be due to deficiencies or limitations of the retrieval algorithm, as no new ash is emitted during this period. We can see a very similar decay and stabilization of ash mass for the AERODYN-rad scenario in green.

**12. P17L375: It would be interesting to know which processes contribute to the removal of ash in the model. I believe the growth term that lead to the removal by sedimentation, what about ash-ice interaction and wet deposition?**

In ICON-ART we account for sedimentation, dry deposition and wet deposition (scavenging by raindrops below clouds). These processes are active for all aerosols for all presented simulation cases, i.e. they are not exclusively linked to the AERODYN development. However, the presented setup of ICON-ART does not account for ash-ice interaction or CCN activation of aerosol yet. Combining aerosol aging with aerosol activation will be subject of future development.
In order to have this information in the manuscript as well, we add the following in l.184:

The removal of aerosols from the atmosphere is modeled by three different processes: sedimentation, dry deposition and wet deposition. In ICON-ART wet deposition describes scavenging by raindrops below clouds.

Furthermore, we discuss the differences of these three mechanisms in l. 377 ff.:
Additionally, we would like to note that the prevailing settling mechanism of aerosol after the Raikoke 2019 eruption for all our simulation scenarios is due to sedimentation. Dry

deposition is only relevant for aerosol near the ground. Wet deposition should also play a minor role during the first days after the eruption, as most of the volcanic ash is emitted above cloud level.

**13. Figure 9: More data are needed to verify the model outputs. e.g. CALIPSO and OMPS.**

As already discussed in our answer to comment 7, we were not able to retrieve additional meaningful OMPS-LP data for plume top heights.
In contrast, there exist several CALIPSO overpasses (which we show in our answer to comment 10). The measurements of total attenuated backscatter at 532 nm on these dates show a signal which can be associated with volcanic aerosol. However, in the scope of this work we were not able to define an objective quantity that allows us determining the volcanic cloud top height in CALIOP measurements. That is the reason why we constrain the comparison between CALIOP and ICON-ART to the more qualitative Fig. 6.

**14. P20L431: I believe that measurement uncertainty from OMPS could be better addressed. The vertical resolution of the instrument is probably near 1-2 km. Could you add the corresponding error bar in figure 8. In addition, I'm pretty confident that additional information on volcanic cloud top height could be found by analyzing additional OMPS data.**

The vertical sampling of OMPS-LP is 1km which gives us +/- 0.5km accuracy in the peak attribution. The remaining uncertainty in the pointing is about 0.2km. The latter is rather systematic. The aerosol retrieval has a vertical resolution of about 3km which smears the peak, however, won't displace it. This is why we estimate the measurement uncertainty with +/- 0.7km.
As suggested, we add an error bar to the OMPS-LP measurement in figure 8.

[Figure]

Figure 8. (a) and (b) Evolution of height of volcanic ash cloud top after the onset of the eruption on 21 June 2019, at 18:00 UTC. The yellow curve represents the no_AERODYN-rad scenario, the green curve AERODYN-rad, the pink one AERODYN-no_rad, and the orange one represents the no_AERODYN-no_rad scenario. Panel (a) shows the ash cloud top of particles in the accumulation mode, (b) of particles in the coarse mode, respectively. The black circle depicts the volcanic cloud top height obtained from OMPS-LP. (c) Mean temperature difference (AERODYN-rad – AERODYN-no_rad) in volcanic ash cloud columns on 23 June 2019, 12:00 UTC. (d) Mean volcanic ash concentration $\bar{\chi}$ for the same model columns as in (c) for AERODYN-rad.

Additionally, we add in l.161:

Due to uncertainties in pointing and vertical sampling we estimate the measurement error with +/- 0.7km.